



Impact of light-absorbing particles on snow albedo darkening and associated radiative forcing
over High Mountain Asia: High resolution WRF-Chem modeling and new satellite observations
Chandan Sarangi[1], Yun Qian[1*], Karl Rittger[2], Kat J. Bormann[3], Ying Liu[1], Hailong Wang[1], Hui
Wan[1], Guangxing Lin[1], and Thomas H Painter[3]
[1]Pacific Northwest National Laboratory, Richland, WA
[2]Institute of Arctic and Alpine Research, Boulder, CO
[3]Jet Propulsion Laboratory, California Institute of Technology, Pasadena, CA
Submitted to the Atmospheric Chemistry and Physics
September 13, 2018
Corresponding Author: yun.qian@pnnl.gov



**Abstract**

20        Light-absorbing particles (LAPs), mainly dust and black carbon, can significantly impact

snowmelt and regional water availability over High Mountain Asia (HMA). In this study, for the
first time, online aerosol-snow interactions enabled and a fully coupled chemistry Weather
Research and Forecasting (WRF-Chem) regional model is used to simulate LAP-induced
radiative forcing on snow surfaces in HMA at relatively high spatial resolution (12 km, WRF-
HR) than previous studies. Simulated macro- and micro-physical properties of the snowpack and
LAP-induced snow darkening are evaluated against new spatially and temporally complete
datasets of snow covered area, grain size, and impurities-induced albedo reduction over HMA. A
WRF-Chem quasi-global simulation with the same configuration as WRF-HR but a coarser
spatial resolution (1 degree, WRF-CR) is also used to illustrate the impact of spatial resolution
on simulations of snow properties and aerosol distribution over HMA. Due to a more realistic
representation of terrain slopes over HMA, the higher resolution model (WRF-HR) shows
significantly better performance in simulating snow area cover, duration of snow cover, snow
albedo and snow grain size over HMA, as well as an evidently better atmospheric aerosol
loading and mean LAPs concentration in snow. However, the differences in albedo reduction
from model and satellite retrievals is large during winter due to associated overestimation in
simulated snow fraction. It is noteworthy that Himalayan snow cover have high magnitudes of
LAP-induced snow albedo reduction (4-8 %) in summer (both from WRF-HR and satellite
estimates), which, induces a snow-mediated radiative forcing of $\sim$ 30-50 W/m$^2$. As a result,
Himalayas (specifically western Himalayas) hold the most vulnerable glaciers and mountain
snowpack to the LAP-induced snow darkening effect within HMA. In summary, coarse spatial
resolution and absence of snow-aerosol interactions over Himalaya cryosphere will result in
significant underestimation of aerosol effect on snow melting and regional hydroclimate.



## 1. Introduction

Light-absorbing aerosol particles (LAPs; airborne dust and black carbon (BC) specks),

can impact on regional water availability over Asia in three ways. Firstly, LAPs can directly
interact with incoming solar radiation and induce thermo-dynamical modifications to synoptic
scale circulations (Hansen et al., 1997 ; Ramanathan et al., 2001; Bond et al., 2013; Lau et al.,
2006; Bollasina et al., 2011; Li et al., 2016). Secondly, acting as cloud condensation nuclei,
changes in concentrations of these particles can lead to microphysical modification of cloud
systems and precipitations (Fan et al., 2016; Li et al., 2016 ; Qian et al., 2009; Sarangi et al.,
2017). Finally, deposition of LAPs in the snowpack can also darken the snow, reduce its surface
albedo and accelerate snow warming and melting (Warren and Wiscombe, 1980; Qian et al.,
2015; Qian et al., 2011; Qian et al., 2009a; Lau et al., 2010; Xu et al., 2009; Hadley and
Kirchstetter, 2012; Dang et al., 2017). Modeling studies have suggested that the LAP-induced
snow darkening mechanism has warming and snow-melting efficacy even greater than that of
greenhouse gases (GHGs) (Hansen and Nazarenko, 2004; Flanner et al., 2007; Qian et al., 2011;
Skiles et al., 2012). To give a perspective, the concentration of just 100 ng of BC in 1 g of
snowpack will reduce the visible-wavelength albedo of grain radius 1000 μm by 10% (Fig. 1b of
Warren, 2013). A chain of positive feedback mechanisms results in such large impact of LAPs
(Qian et al., 2015). Initially, as snow starts to melt, the concentration of LAPs in snowpack
increases because a portion of LAPs accumulate at the surface of the snowpack instead of getting
washed away with meltwater (Conway et al., 1996; Flanner et al., 2007; Doherty et al., 2010).
This increase in LAP concentration leads to enhanced warming of the snowpack and thereby
increases the effective snow grain size, which further lowers snow albedo (Warren and
Wiscombe, 1980; Hadley and Kirchstetter, 2012). Nonetheless, at higher concentrations, grain



sizes can again be reduced due to the loss of mass from surface layers with the intense melting
(Painter et al., 2013).  As this process continues, sufficient snow melt occurs to expose the darker
underlying surface, leading to enhanced warming and snow ablation commonly-known as "snow
albedo feedback" (Warren and Wiscombe, 1980; Hansen and Nazarenko, 2004; Flanner et al.,
2007; Qian et al., 2015). In turn, this earlier loss of snow cover induces surface warming and
perturbing regional circulations (Hansen and Nazarenko, 2004; Lau et al., 2010; Qian et al.,
2011). This LAP-induced modification of snow albedo feedback is identified as one of major
forcing agents affecting climate change with a high level of uncertainty (IPCC, 2013).

High Mountain Asia (HMA) includes the Tibetan plateau, central Asian mountains and

the Himalaya cryosphere. It holds the largest glacial cover (~9500 glaciers) outside the polar
region (Dyurgerov, 2001).  Observations revealed that a historical decadal increase in the surface
air temperature over HMA in a range of 0.6-1.8 °C (Shrestha et al., 1999;Wang et al., 2008), and
the warming is faster over higher elevations (> 4000 m) in the last three decades (Xu et al.,
2009b; Ghatak et al., 2014). The Himalaya glacier area has cumulatively decreased by ~16%
during the period 1962 to 2004 (Kulkarni et al., 2010) and the spring snow cover is decreasing at
a decadal rate of ~0.8 million km$^2$ during the last 50 years (Brown and Robinson, 2011). The
average retreat rate on the north slope of Mount Everest is as high as 5.5–9.5 m y$^{-1}$ (Ren et al.,
2006). The Himalaya cryosphere contributes to the stream flow in Indus and Ganges river
systems by ~ 50 % and ~10-30%, respectively (Khan et al., 2017). Warming and glacier retreat
over the Himalaya cryosphere have a great potential to impact the fresh water availability for
about 700 million people, modify regional hydrology, and disturb the agrarian economy of all
South Asian countries (Bolch et al., 2012;Immerzeel et al., 2010;Kaser et al., 2010;Singh and



Bengtsson, 2004;Barnett et al., 2005;Yao et al., 2007). Therefore, it is critical to disentangle the
factors contributing to glacier retreat and snow melt over HMA.
Regional warming due to increasing greenhouse gases (Ren and Karoly, 2006) has been reported
as the primary cause of the high rate of warming and glacier retreat over HMA. However, in the
last decade, advancement in remote sensing and availability of measurements from several field
campaigns suggest that the contribution of LAP loading (in the atmosphere) to the warming and
glacier melting over HMA is probably greater than previously believed (Ramanathan et al.,
2007; Prasad et al., 2009; Menon et al., 2010). Continuous observations over the Nepal Climate
Observatory Pyramid (NCO-P) facility located at 5079 m a.s.l. in the southern foothills of Mt.
Everest revealed very high concentrations of black carbon (Marcq et al., 2010) and desert dust
(Bonasoni et al., 2010) especially in spring from Indo-Gangetic plains. Atmospheric LAPs are
scavenged to the snow/ice surface by dry and wet deposition and cause measurable snow
darkening and melting (Gautam et al., 2013; Yasunari et al., 2010b;Yasunari et al., 2013; Nair et
al., 2013; Ménégoz et al., 2014; Ming et al., 2008;Flanner and Zender, 2005). Thus, LAP
deposited in snow and associated snow darkening has been suggested as a key factor to the early
snowmelt and rapid glacier retreat over HMA (Yasunari et al., 2010; Ming et al., 2008; Xu et al.,
2009a; Flanner et al., 2007; Qian et al., 2011).
While previous studies have underlined the significance of LAP-deposition in snow over
HMA, the estimation of LAP-induced snow darkening and associated radiative forcing is still
highly uncertain (Qian et al., 2015). Many of these studies used online global model simulations
at coarse spatial resolutions of ~50-150 km (Flanner and Zender, 2005; Ming et al., 2008; Qian
et al., 2011). Others used offline simulation of the snow albedo effect using measured
concentrations of deposited LAP in surface snow or estimated from atmospheric loading and ice



cores (Yasunari et al., 2013; Nair et al., 2013; Wang et al., 2015). The complex terrain of HMA,
seasonal snowfall and near surface air circulation are not well resolved by coarse global climate
models (Kopacz et al., 2011; Ménégoz et al., 2014). Similarly, offline estimations are limited in
scope because they are site specific and are based on simplified assumptions about deposition
rates. Ideally, online high resolution simulations allowing for LAP-snow interactions should
facilitate a more realistic understanding of LAP deposition to snow and LAP-induced snow
darkening effect in terms of both magnitude and spatial variability over HMA.

In this study, a modified version of the online chemistry coupled with Weather Research

and Forecasting regional model (WRF-Chem v3.5.1), which, is then fully coupled with SNICAR
(SNow, ICe, and Aerosol Radiative) model, is used to perform first-ever high resolution (12 km)
simulation over the HMA region for the water year 2013-14 (October 1, 2013 to September 30,
2014).  Satellite observations of snow properties like snow albedo, grain size, and LAP-induced
snow darkening from MODSCAG and MODDRFS retrievals are used for evaluation (Painter et
al, 2009; 2012). The main objective of this study is to evaluate the skill of high resolution WRF-
Chem model in simulating properties of snowpack, aerosol distribution, LAP in snow and LAP-
induced snow darkening over HMA using spatially and temporally complete (STC) remotely
sensed snow surface properties (SSP) from MODIS (Dozier et al, 2008; Rittger et al, 2016). Our
second objective is to demonstrate the benefit for aerosol and snow distributions in high
resolution runs by comparing to a coarse gridded quasi-global model simulation over HMA. This
quasi-global simulation is run with the same WRF-Chem configuration but at 1 degree spatial
resolution. Finally, the spatiotemporal variation of simulated LAP deposition, snow albedo
darkening and snow mediated LAP radiative forcing (LAPRF) over HMA are discussed. The



model details and datasets used are described in Section 2. Results and discussions are presented
in Section 3 followed by conclusions in Section 4.
**2: Model simulations and observational datasets**

Below, we provide details on the aerosol module used in WRF-Chem, interactive

coupling with aerosol and SNICAR via land model, and the model setup for both 12 km and 1
degree resolution runs. The details for the remote sensing observations that are used to evaluate
the models are also provided.
**2.1: Coupled WRF-Chem-CLM-SNICAR Model description**

The WRF-Chem simulation is performed at 12 km × 12 km horizontal resolution

(hereafter refereed as WRF-HR) with 210 × 150 grid cells (64–89°E, 23–40°N) (Figure 1) and
35 vertical layers. The simulation was conducted from 20[th] September, 2013 to 30[th] September,
2014, to provide one year of results (following a 10-day model spin-up). ERA-interim reanalysis
data at 0.7° horizontal resolution and 6 h temporal intervals are used for meteorological initial
and lateral boundary conditions. The simulation is re-initialized every 4th day to prevent the drift
of model meteorology. Model physics options used are the MYJ (Mellor–Yamada– Janjic)
planetary boundary layer scheme, Morrison 2-moment microphysics scheme, community land
model (CLM), Kain-Fritsch cumulus scheme and Rapid Radiative Transfer Model for GCMs
(RRTMG) for longwave and shortwave radiation schemes.

The CBM-Z (carbon bond mechanism) photochemical mechanism (Zaveri and Peters,

1999) coupled with eight bin MOSAIC (Model for Simulating Aerosol Interactions and
Chemistry) aerosol model (Zaveri et al., 2008) is used. This is the most sophisticated aerosol
module available for the WRF-Chem model. The sectional approach with eight discrete size bins



is used to represent the size distributions of all the major aerosol components (including sulfate,
nitrate, ammonium, black carbon (BC), organic carbon (OC), sea salt, and mineral dust) in the
model. The processes of nucleation, condensation, coagulation, aqueous phase chemistry, and
water uptake by aerosols in each bin size are included in the MOSAIC module. Dry deposition of
aerosol mass and number is simulated by including both diffusion and gravitational effects as per
Binkowski and Shankar (1995). Wet removal of aerosols follow Easter et al. (2004) and
Chapman et al. (2009) and includes grid resolved impaction and interception processes for both
in-cloud (rainout) and below-cloud (washout) aerosol removal. Processes involved in convective
transport and wet removal of aerosols by cumulus clouds are described in Zhao et al. (2013).
Anthropogenic emissions used in our study is at $0.5^O \times 0.5^O$ horizontal resolution and are
taken from the NASA INTEX-B mission Asian emission inventory for year 2006 (Zhang et al.,
2009). Biomass burning emissions at $0.5° \times 0.5°$ horizontal resolution for the water year 2013-14
are obtained from the Global Fire Emissions Database, Version 3 (GFEDv3) (Van Der Werf et
al., 2010), which are vertically distributed in our simulation using the injection heights
prescribed by Dentener et al. (2006) for the Aerosol Inter Comparison project (AeroCom). Sea
salt and dust emissions follow Zhao et al. (2014). Dust surface emission fluxes are calculated
with the Georgia Institute of Technology-Goddard Global Ozone Chemistry Aerosol Radiation
and Transport (GOCART) dust emission scheme (Ginoux et al., 2001), and emitted into the eight
MOSAIC size bins with respective mass fractions of $10^{-6}$ , $10^{-4}$ , 0.02, 0.2, 1.5, 6, 26, and 45%.
Aerosol optical properties are computed as a function of wavelength for each model grid
cell. The Optical Properties of Aerosols and Clouds (OPAC) data set (Hess et al., 1998) is used
for the shortwave (SW) and longwave (LW) refractive indices of aerosols and a complex
refractive index of aerosols (assuming internal mixture) is calculated by volume averaging for



each chemical constituent of aerosols for each bin. A spectrally-invariant value of $1.53 \pm 0.003i$
is used for the SW complex refractive index of dust. Fast et al. (2006) and Barnard et al. (2010)
provide detailed descriptions of the computation of aerosol optical properties such as extinction
coefficient, single scattering albedo (SSA), and asymmetry factor in WRF-Chem. Following
Zhao et al. (2011) and Zhao et al. (2013a), aerosol radiative feedback is coupled with the Rapid
Radiative Transfer Model (RRTMG) (Mlawer et al., 1997) and the direct radiative forcing of
individual aerosol species in the atmosphere are diagnosed. Aerosol–cloud interactions are
included in the model following Gustafson et al. (2007).

The increasingly used Snow, Ice, and Aerosol Radiation (SNICAR) model simulates the

snow properties and associated radiative heating rates of multilayer snow packs (Flanner and
Zender, 2005; Flanner et al., 2009,  2012 and 2007). Fundamentally, it employs the snow albedo
theory (parameterization) based on Warren and Wiscombe (1980) and the two-stream radiative
approximation for multilayers from Toon et al. (1989). SNICAR can also simulate aerosol
radiative effect in snow for studying the LAP heating and snow aging (Flanner et al., 2007).
Recently, laboratory and site measurements are used to validate the SNICAR simulated change
of snow albedo for a given BC concentration in snow (Hadley and Kirchstetter, 2012; Brandt et
al., 2011). For radiative transfer calculations, SNICAR defines layers matching with the five
thermal layers in community land model (CLM) that vertically resolve the snow densification
and meltwater transport (Oleson et al., 2010). In WRF-Chem-SNICAR coupled model, BC and
dust deposition on snow is calculated in a prognostic approach through dry and wet deposition
processes. BC in snow can be represented as externally and internally mixed with precipitation
hydrometeors depending on the removal mechanism involved, but dust is considered to only mix
externally with snow grains (following Flanner et al., 2012). SNICAR in WRF-Chem simulates



four tracers of dust based on size (with diameters of 0.1–1, 1–2.5, 2.5–5, and 5–10 μm) and two
tracers of BC (externally and internally mixed BC with 0.2 μm dry diameter) in snow. The
MOSAIC aerosol model simulates dust in the atmosphere with eight size bins (0.039–0.078,
0.078–0.156, 0.156–0.312, 0.312–0.625, 0.625 - 1.25, 1.25–2.5, 2.5–5.0, and 5.0–10.0 μm in dry
diameter). The first 4 bins are coupled with the smallest bin of dust particles in SNICAR. While
the next two MOSAIC bins (5th and 6th) map into the second bin of SNICAR, the 7th and 8th
MOSAIC dust bins correspond to the third and fourth SNICAR dust bins (Zhao et al., 2014),
respectively. Deposition of LAPs to snow in SNICAR are immediately mixed in the CLM
surface snow layer (< 3 cm). CLM adds excess water in the layer above to the layer beneath
during melting. The scavenging of aerosols in snow by meltwater is assumed to be proportional
to its mass mixing ratio of the meltwater multiplied by a scavenging factor. Scavenging factors
for externally mixed BC and internally mixed BC are assumed to be 0.03 and 0.2, respectively,
and 0.02, 0.02, 0.01, and 0.01 for the four dust bins (all externally mixed). Although these
scavenging factors are comparable to observations (Doherty et al., 2013), the scavenging ratios
can be highly heterogeneous and introduce high uncertainty into the estimation of LAP
concentrations in snow (Flanner et al., 2012; Qian et al., 2014). More detailed description about
the aerosol deposition and mixing processes, computation of optical properties of snow and
LAPs in WRF-Chem-CLM-SNICAR coupling can be found in Zhao et al.(2014) and Flanner et
al.(2012).

Configured in the way similar to the WRF-HR, a coarse (1° × 1°) gridded WRF-Chem

simulation is also performed using a quasi-global model (hereafter referred as WRF-CR) with
360 × 130 grid cells (180° W–180° E, 60° S–70° N). Periodic boundary conditions are used in
the zonal direction. Reanalysis of the TROpospheric (RETRO) anthropogenic emissions for the



year 2010 (ftp://ftp.retro.enes.org/ pub/emissions/aggregated/anthro/0.5x0.5/) is used for
anthropogenic aerosol and precursor gas emissions in the coarse gridded quasi-global WRF-
Chem simulation except for Asia and the United States. INTEX-B anthropogenic emissions
(Zhang et al., 2009) and US National Emission Inventory are used for Asia and the U.S.,
respectively. Emissions of biomass burning aerosols, sea salt, and dust are treated in the same
way as described above for the WRF-HR simulation. More details about the quasi-global WRF-
Chem simulation can be found in (Zhao et al., 2013b);Hu et al., 2016). Chemical initial and
boundary conditions to the WRF-HR simulation are provided by this quasi-global WRF-CR runs
for the same time period to include long-range transported chemical species.
**2.2: Aerosol Optical Depth (AOD) dataset**
The aerosol robotic network (AERONET – https://aeronet.gsfc.nasa.gov) is a global
network of ground based remote sensing stations that provides quality-controlled measurements
of AOD with uncertainties ~0.01 under clearsky conditions over India (Holben et al., 1998;
Dubovik et al., 2000). CIMEL Sun scanning spectral radiometers are used to measure direct Sun
radiance at eight spectral channels (340, 380, 440, 500, 675, 870, 940, and 1020 nm) and
measure spectral columnar AOD (Holben et al., 1998). AERONET provides measurements at
~15 min temporal resolution from sunrise to sunset.
Skyradiometer Network (Skynet) is another global network of ground based spectral
scanning radiometer (POM-01L, Prede, Japan) stations that provides quality-controlled
measurements of AOD (Nakajima et al., 1996). With an automatic sun scanner and sensor, it
measures sky irradiance in five wavelengths i.e. 400, 500, 675, 870, and 1020 nm. The measured
monochromatic irradiance data is processed by using Skyrad.Pack version 4.2 software.
Calibration of the Sky radiometer is carried out on a monthly basis (http://atmos3.cr.chiba-





u.jp/skynet/data.html). Details of the instrumentation and software protocol can be found in
Campanelli et al. (2007) and Ningombam et al., (2015).  In this study, we have also used AOD
measurements at 500 nm over MERAK, a high altitude Skynet station in Himalaya for water
year 2013-14.

The MODerate resolution Imaging SpectroRadiometer (MODIS) instrument onboard the

NASA AQUA satellite provides global coverage of daily radiance observations (at 1330 LT) in
36 spectral channels. Over North India, Tripathi et al., (2005) has shown that MODIS
observations correlate well with ground based measurements. For the evaluation of model
simulated AOD, 1° gridded Level 3 AOD estimates (collection 6) at 0.55 µm wavelength
obtained from the MODIS instrument are used during water year 2013-14. However, the MODIS
land aerosol algorithm uses a dark target approach (Levy et al., 2007), which, is known to have
large uncertainties over arid and mountainous surfaces (Levy et al., 2010).
**2.3: Spatially and temporally complete MODSCAG and MODDRFS retrievals**

Subpixel snow-covered area and snow grain size are retrieved from MODIS-observed

surface reflectance data using the physically based MODIS Snow-Covered Area and Grain size
(MODSCAG) (Painter et al., 2009) algorithm. In each snow covered pixel, MODSCAG
attributes a fractional snow-covered area and grain size using spectral mixture analysis to
determine proportion of the pixel that is snow and is not snow. MODSCAG is more accurately
identifies snow cover throughout the year than the widely used MODIS snow product:
MOD10A1 (Rittger et al., 2013). The MODSCAG snow-mapping algorithm for fraction of snow
covered area has an uncertainty of ~ 5 % (Rittger et al., 2013). The current study incorporates
pixel level snow cover area and snow grain size from MODSCAG over the HMA region to
evaluate snow pack simulation and LAP-induced albedo reduction. Further, MODIS Dust



Radiative Forcing in Snow (MODDRFS) model (Painter et al., 2012) is used to determine the
LAP-induced albedo reduction over HMA. MODDRFS uses spectral reflectance differences
between the measured snow spectral albedo and the modeled clean snow spectral albedo. The
pixel level clean snow spectrum corresponding to MODSCAG retrieved snow grain sizes is
calculated using discrete ordinate radiative transfer solutions for visible wavelengths and solar
zenith angles. Coupled, these products provide the determination of snow albedo for the
fractional snow cover with LAP inclusion. Reflectance inputs to MODSCAG and MODDRFS
are degraded by cloud cover, off-nadir views, and data errors, but can be filtered in time and
space to improve data quality and consistency. Our method for spatially cleaning and filling
(Dozier et al., 2008;Rittger et al., 2016) combines noise filtering, snow/cloud interpolation and
smoothing to improve the daily estimates snow surface properties (SSP). Using remotely sensed
forest height maps (Simard et al., 2011) and MODSCAG vegetation fraction, we adjust the
satellite viewable snow cover to account for snow under tree canopy (Rittger et al., 2016).  We
weight the observations based on satellite viewing angle that varies from 0 to 65 degree with
larger uncertainties in off-nadir views (Dozier et al., 2008). The result is a set of spatially and
temporally complete (STC) SSPs. Use of these products in an energy balance model to estimate
snow water equivalent based on reconstruction produced more accurate snow cover than the
Snow Data Assimilation System (SNODAS) or an interpolation of observations from snow
pillows (Bair et al., 2016). In this study we use STC versions of MODSCAG and MODDRFS
when comparing our WRF model output. The incomplete remotely sensed would be difficult to
use given the gaps in data and uncertainties related to viewing angle (Dozier et al., 2008).
Hereafter, the use of MODSCAG and MODDRFS terms will invariably refer to these STC-
MODSCAG and STC-MODDRFS products.





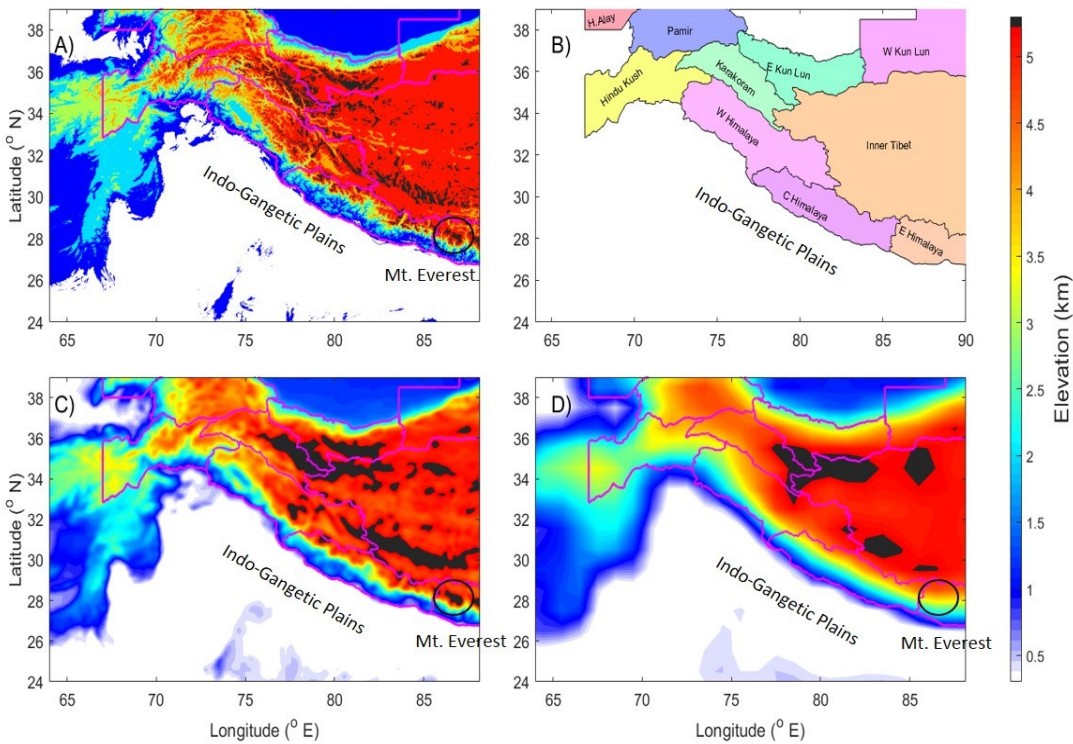

Figure 1: Panel A illustrates the terrain elevation at 1 km resolution from ETOPO1 dataset. Panel
B shows glacier classification over HMA region used in this study following Randolph Glacier
Inventory. Panels C and D illustrate terrain representation in WRF-HR run (12 km) and WRF-
CR (1 degree) runs, respectively. For reference, Mt. Everest (shown as black circle in Figure 1C)
is distinctly represented in Panel A and C, but not in Panel B.

**2.4: Variation in terrain representation in WRF-HR and WRF-CR**
Figure 1A illustrates the variations in terrain height over HMA at a resolution of 1 arc
using ETOPO1 Global Relief Model, a publicly available global topographical dataset (Amante
and Eakins, 2009). It clearly shows the enormous relief in terrain as we move from the Indo
Gangetic plains (IGP) to the crest of the Himalaya and into the Tibetan Plateau (TP). The
majority of HMA is above 4 km altitude with many Himalaya peaks at an altitude higher than 6
km. Figure 1B illustrates the mountain ranges and glaciers classified as per the Randolph Glacier



Inventory in the Fifth Assessment Report of the Intergovernmental Panel on Climate Change
(Pfeffer et al., 2014). Specifically, Pamirs, Hindu Kush, Karakoram, Kunlun, and Himalaya hold
the most number of glaciers in HMA. Figures 1C and 1D illustrate the representation of terrain
elevation in WRF-HR and WRF-CR, respectively. Compared to Figure 1A, location of mountain
peaks (altitude > 5.5 km) are better represented in WRF-HR compared to WRF-CR, as is
particularly evident over the Karakoram, Kunlun, and Himalaya ranges. Moreover, the steep rise
in elevation between IGP and TP is also well represented by WRF-HR, whereas it is more
gradual in WRF-CR.
**2.5: Methodology**

Simulation of the snow macro- and micro-physical properties, aerosol loading and LAP

in snow concentration from WRF-HR, WRF-CR and observational estimates (datasets described
above) over HMA are compared in Section 3.1 and Section 3.2. In Section 3.3, the WRF-HR
simulated LAP-induced snow albedo reduction values over HMA is compared with
corresponding MODIS satellite based STC-MODSCAG and STC-MODDRFS. Lastly, a
discussion on the high resolution model simulated LAP-induced radiative forcing estimates over
HMA is also presented in context to previous studies and other atmospheric forcing.

The simulated fractional snow covered area (fSCA), duration of snow cover over a grid

in terms of number of snow cover days (NSD), snow albedo ($\alpha$) and snow grain sizes (SGS) and
LAP-induced snow albedo darkening ($\Delta\alpha$) for midday (1000 -1400 LT) conditions from both the
WRF models are compared with corresponding STC-MODSCAG and STC-MODDRFS
retrievals over HMA. The number of snow cover days (NSD: defined as days having fSCA
values $\geq 0.01$) during water year 2013-14 is determined over each grid from STC-MODSCAG
and both model runs. They are compared with corresponding values from STC-MODSCAG



retrievals, which, are observed during Terra overpasses at 10:30 LT. We have used a window
from 10:00 LT to 16:00 LT for representing midday averages of modelled variables to
incorporate the variability due to differences in timing (between model and real scenario) of
weather conditions like precipitation and clouds. In addition, the change in snow albedo during
10:00 - 14:00 LT is < 0.01 (Bair et al., 2017), which, is low compared to other model physics-
and data retrieval related uncertainties. The WRF-CR simulated variables and STC-MODSCAG
and STC-MODDRFS retrievals are gridded to the resolution of WRF-HR (12 km) for ease of
comparison. We have compared annual mean values as well as seasonal mean values for winter
(December - February) and summer (April - June) season, separately. We have not considered
the monsoon period in our analyses because the snow cover during the monsoon is negligible
(except in glaciated regions at high altitudes) relative to other months (Figure S1). To evaluate
spatial heterogeneity in our model, the seasonal and annual distribution of these variables are
calculated separately for each sub region (shown in Figure 1) within HMA. In addition, to gain
an understanding of the extent of temporal variability present in LAP-induced effects, we have
also presented daily midday variation in LAP-induced snow darkening and LAP-induced
radiative forcing at surface over Chotta Shingri glacier region (32.1-32.35 °N, 77.4-77.7 °E)
located in the Chandra–Bhaga river basin of Lahaul valley, Pir Panjal range, in Western
Himalayas. It is an accessible and representative site for glacier mass balance studies in western
Himalayas. Chotta Shigri glacier has a cumulative glaciological mass loss of −6.72 m w.e.
between 2002 and 2014 (Azam et al., 2016).

The simulated aerosol optical depth (AOD) is compared with available in-situ

observations (described in Section 2.2). Here, quality assured (Level 2) midday (1000 to 1400
LT) averages of AOD (550 nm) at seven AERONET stations (Lahore, Jaipur, Kanpur, Gandhi



college, Kathmandu and CAS) and one SkyNet site within our study region are used to evaluate
the simulated AOD values. Further, the simulated distribution of LAP concentration in snow at a
few sites is compared with field measurements. Only a few field measurements of concentration
of BC ($LAP_{BC}$) and dust particles ($LAP_{dust}$) in the snow surface or the surface layer are available
over glaciated regions within our study domain. In this study, measurements of $LAP_{BC}$ over
Muztagh Ata in eastern slopes of Pamirs (Xu et al., 2006), Uttaranchal region of W. Himalayas
(Svensson et al., 2018), East Rongbuk at 6.4 km altitude (Ming et al., 2012; Ming et al., 2008;
Xu et al., 2009a) and composite of recent in-situ measurements from various studies near the
NCO-pyramid site in Nepal at 5-6 km altitude (Kaspari et al., 2014; Yasunari et al., 2013; Jacobi
et al., 2015; Ginot et al., 2014) is used. Similarly, the point measurements used for evaluating
$LAP_{dust}$ are over Abramov glacier in western slopes of Pamirs (Schmale et al., 2017), Muztagh
Ata in eastern slopes of Pamirs (Wake et al., 1994) , East Rongbuk (Ming et al., 2012) and near
NCO-pyramid station (Ginot et al., 2014), respectively.
**3: Results and Discussions**
**3.1: Snow physical, microphysical and optical properties**
The largest values of region-averaged annual mean fSCA within HMA are observed in both the
satellite retrievals and the model runs over the Karakoram region (mean=0.45) followed by
Pamirs, Himalayas and Hindu Kush in the HMA region (Figures 2A and 2B). In comparison, the
fSCA over Kunlun and TP are lower (<0.3), but, pockets of very high fSCA (~0.7) are visible
over the west Kunlun ranges (Figure 2A). The annual mean fSCA values and the fine spatial
variability are well simulated by WRF-HR (Figure 2B) over the entire HMA region. Of
exception are simulations over the Karakoram, where WRF-HR overestimates annual mean
fSCA, however, the distribution of annual mean fSCA from WRF-HR and STC-MODSCAG



agree in all the sub regions (Figure 2D). This observation is largely valid also for summer
months (Figure 2F). But, significant overestimation in distribution of fSCA (by >0.2) during
winter is present over Pamirs, Karakoram, W. Himalayas, TP and Kunlun region (Figures 2E).
STC-MODSCAG retrievals illustrate that the Pamirs (NSD=230 days) and Karakoram
(NSD=270 days) ranges remain snow covered for 7-9 months of the year (Figure 3A). Similarly,
the grids in Hindu Kush (NSD=194), W. Himalayas (NSD=189 days) and C. Himalayas
(NSD=191 days) are snow covered for ~ 6-7 months. Mountains in E. Himalayas (NSD=142
days) remain snow covered for only 4-5 months of the year. The distribution of annual NSD
values simulated by WRF-HR in each sub region is close to STC-MODSCAG values (Figure
3D).  Also, the spatial distribution and magnitude of simulated NSD by WRF-HR is similar to
that from STC-MODSCAG for different seasons, separately (Figure S2). Thus, overestimation of
annual mean fSCA in WRF-HR during winter is not due to mere averaging error associated with
underestimation in simulated NSD during winter (Figure S2).



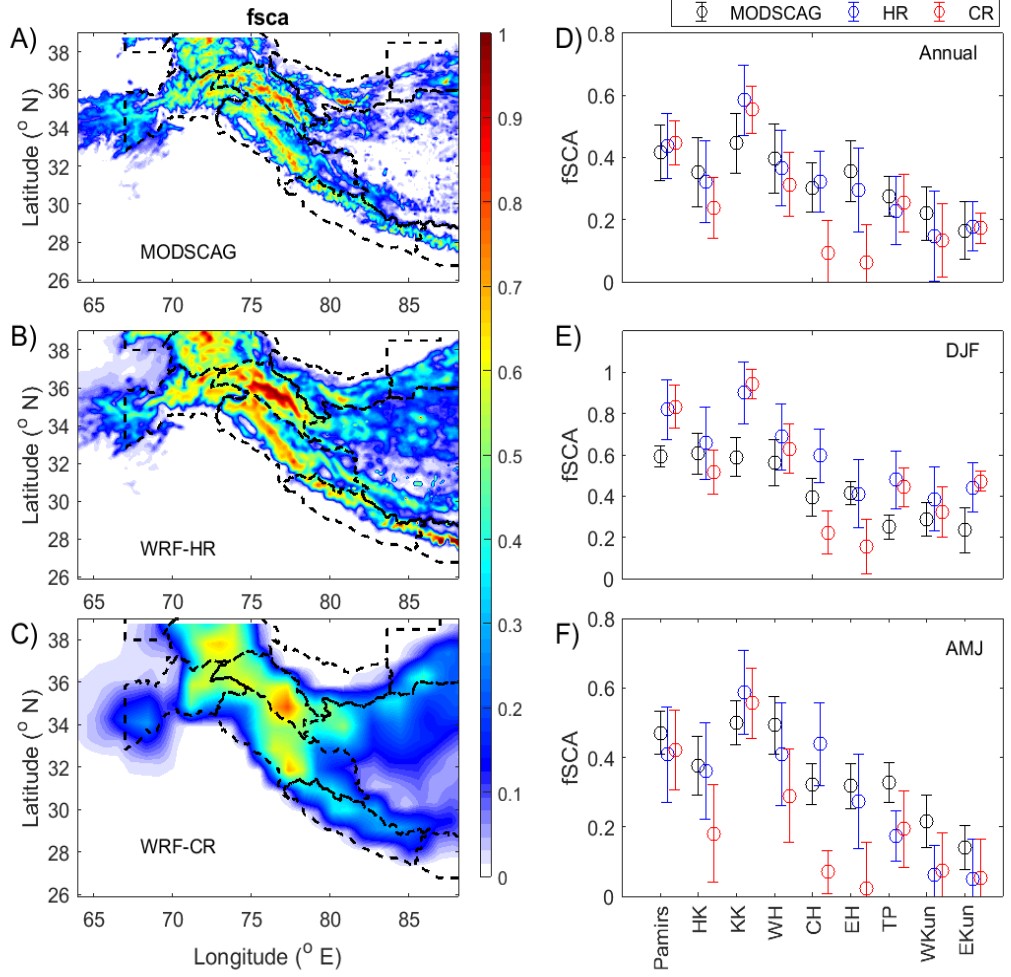


Figure 2: Spatial distribution of annual mean snow cover fraction (fSCA) during midday (1000-
1400 LT) for water year 2013-14 from A) STC-MODSCAG retrievals, B) simulated values from
WRF-HR and C) WRF-CR simulations. Panels D-F illustrate the distribution of midday mean
fSCA over each subregion identified by glacier classification following Randolph Glacier
Inventory (X-axis). The circle and vertical legs represent mean±standard deviation over each
region for D) entire year, E) winter (December - February) and F) summer (April - June) season,
separately. Here, Hindu Kush, Karakoram, W.Himalayas, C.Himalayas, E.Himalayas, Tibetan
Plateau, West Kunlun and East Kunlun regions are abbreviated as HK, KK, WH, CH, EH, TP,
WKun and EKun, respectively.






We also calculated the number of days of snow cover with low fSCA (<0.5; Figure 3B)
and high fSCA (>0.5; Figure 3C) values, separately. The grids in Kunlun, Northern slope of
Karakoram, eastern slope of Pamirs and TP region are dominated by snow cover of relatively
low fSCA for most of their snow cover duration (Figure 3B). But, grids in Hindu Kush,
Himalayas and southern slopes of Karakoram are generally covered with high fSCA values
throughout the year (Figure 3C). The distribution of simulated NSD values over each sub region
for low and high fSCA scenario is shown in Figures 3E and 3F, respectively. WRF-HR can well
simulate the NSD over grids with high fSCA (Figure 3F) but significantly underestimates NSD
over grids with low snow cover (Figure 3E). Note that the regions dominated by low annual
fSCA in this water year are actually the same regions where WRF-HR simulated fSCA values
are being overestimated in winter (Figure 2E). Thus, simulation of fewer number of days with
low fSCA (and vice versa) in WRF-HR might also be contributing partially to the overestimation
of winter fSCA simulated in WRF-HR compared to STC-MODSCAG. Interestingly, WRF-CR
simulated NSD values for low fSCA case is in better agreement with STC-MODSCAG  values
(Figure 3E). Winter mean distribution of WRF-CR simulated fSCA over Kunlun, W.Himalaya
and TP region better match STC-MODSCAG values than the corresponding WRF-HR simulated
fSCA values (Figure 2F). It is noteworthy that these subregions (which are dominated by low
fSCA grids) receive snowfall from western disturbances during winter months. The cloud cover
associated with the western disturbances over these sub regions are extensive in winter which
also introduces uncertainty in MODSCAG retrievals and STC processing and contributes to the
differences between WRF-HR and MODSCAG in fSCA.



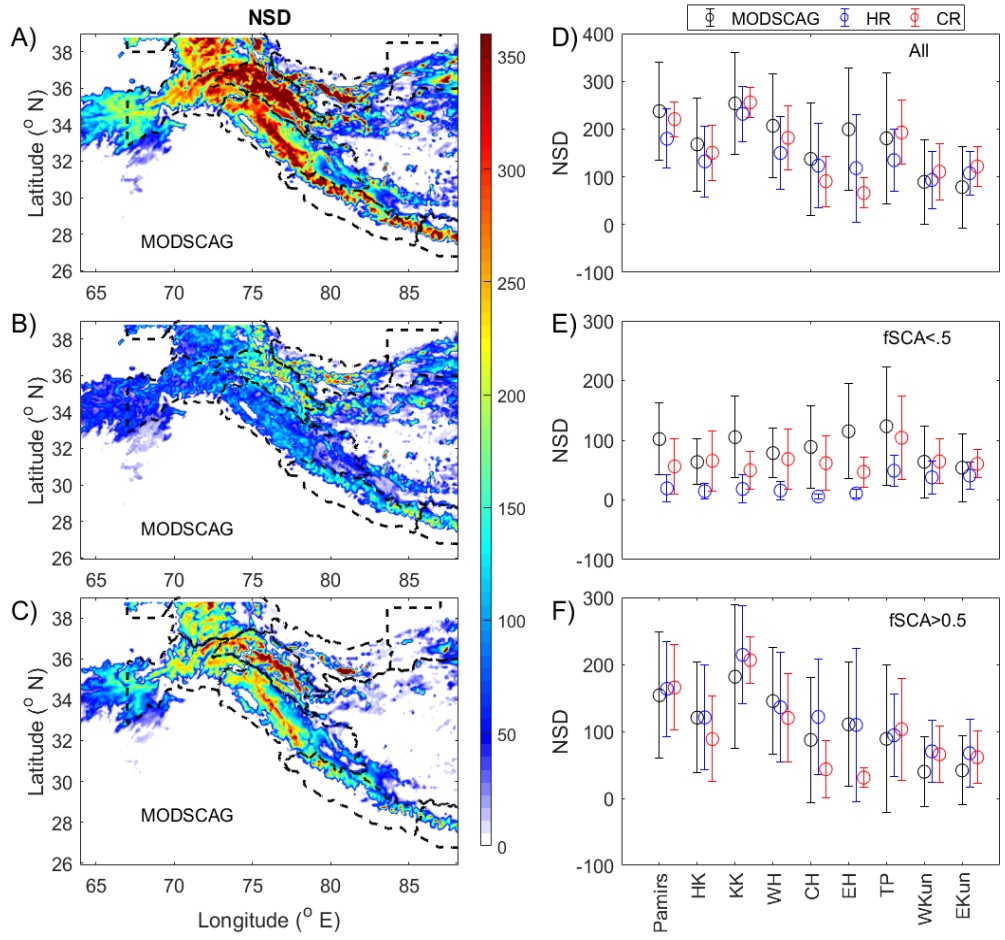


Figure 3: Spatial distribution of snow duration in terms of NSD from A) STC-MODSCAG
retrievals. Panel B and C are similar to Panel A, but, shows number of days when fSCA is below
and above 0.5 over each grid, respectively. Panels D illustrate the distribution of NSD over each
sub region identified by glacier classification following Randolph Glacier Inventory. The circle
and vertical legs represent mean ± standard deviation over each region for entire year. Here,
Hindu Kush, Karakoram, W.Himalayas, C.Himalayas, E.Himalayas, Tibetan Plateau, West
Kunlun and East Kunlun regions are abbreviated as HK, KK, WH, CH, EH, TP, WKun and
EKun, respectively. Panel E and F are similar to Panel D, but, for NSD corresponding to fSCA
values below and above 0.5, respectively.


Comparison between performance of WRF-CR and WRF-HR for fSCA clearly show
significant improvements in the WRF-HR simulations over the Hindu Kush and Himalayan



ranges (Figure 2D). For instance, the simulated annual mean fSCA in WRF-CR around Mt.
Everest (shown as black circle in Figure 1) is less than 0.1 (Figure 2C). This is contrary to the
high fSCA values observed at Mt. Everest (0.7 in Figures 2A) and simulated by WRF-HR (0.7 in
Figures 2B). Moreover, the improvement is present in both winter and summer months
indicating it's independence from meteorological variations (Figures 2E and 2F). Analysis of
NSD values indicate that the snow cover duration in WRF-HR also improved significantly over
these slopes (Figure S3) irrespective of the season. Note that WRF-CR underestimates the snow
duration over Hindu Kush and Himalayas by ~2-6 months (Figure S3) and the spatial location of
grids with very high annual mean fSCA values (mountain ranges) improved in WRF-HR
compared to the STC-MODSCAG data (Figures 2A-C). The observed improvement in fSCA and
NSD simulation over the slopes of Himalaya and Hindu Kush can be attributed to better terrain
representation in WRF-HR.

Next, the simulated microphysical properties of the snow pack are evaluated against the

remote sensing retrievals. Spatial patterns in annual mean SGS from STC-MODSCAG are
similar to that seen in fSCA with highest values over the Karakoram and Himalayan ranges
(Figures 4A) corresponding to the highest elevations and likely the coldest temperatures
hindering snow grain growth. This spatial distribution of annual mean SGS values is well
simulated in WRF-HR runs (Figure 4B). But, the annual mean values are largely overestimated
by 30-50 micron (Figure 4D) relative to STC-MODSCAG.  The seasonal distribution of region-
segregated SGS values from WRF-HR also compares well with that from STC-MODSCAG
retrievals (Figure 4E and 4F). Simulated annual mean SGS from WRF-CR (Figure 4C) lack the
fine spatial variability seen in STC-MODSCAG and WRF-HR. Moreover, the SGS estimates are
largely underestimated (by up to 100 microns) by WRF-CR specifically over grids in central and



eastern Himalayas, TP and Kunlun ranges (Figure 4D). The large underestimation of SGS from
WRF-CR and overestimation of SGS from WRF-HR is present for both summer and winter
months (not shown). The overestimation of SGS from WRF-HR values corroborate well with the
finding that the simulated fSCA distribution from WRF-HR is largely skewed towards higher
values (Figure 3). Similarly, the unrealistically low mean values of SGS from WRF-CR over
Himalayas, TP and Kunlun ranges are consistent with the underestimation of fSCA and NSD
values over these regions (Figure 2 and 3). While, SGS retrievals from STC-MODSCAG are
based on observed surface reflectance, the modeled SGS is calculated from simulated snow mass
in top model layer in the grid. Hence, improvement in simulation of fSCA and NSD in high
resolution WRF-HR runs also caused the SGS values from WRF-HR to be closer to STC-
MODSCAG retrievals than SGS from WRF-CR runs. It is worth noting that the presence of
cloud cover influences STC-MODSCAG retrievals of SGS towards smaller grain sizes if clouds
are misidentified as snow. This systematic error could also contribute to the SGS differences
between WRF-HR and STC-MODSCAG estimates.






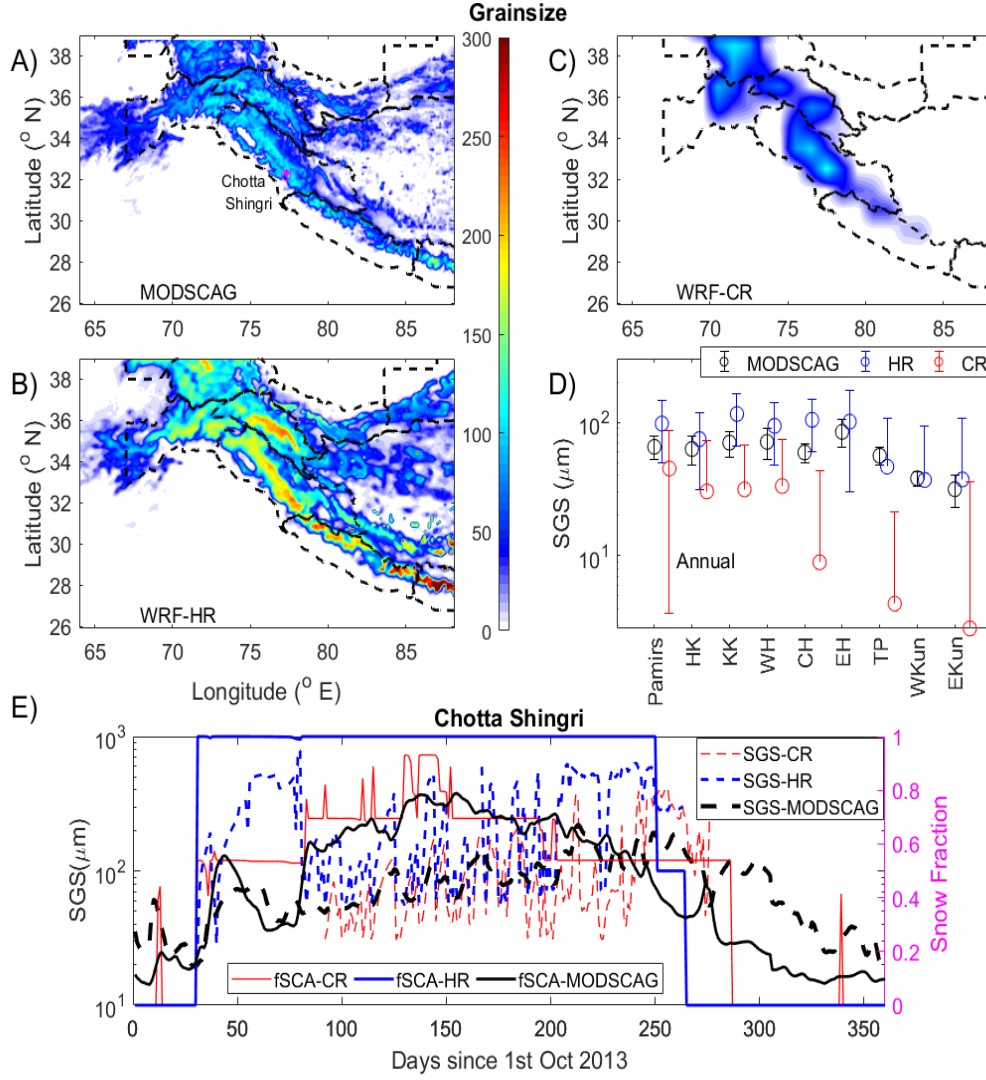

Figure 4: Spatial distribution of annual mean snow cover fraction (fSCA) during midday (1000-1400 LT) for water year 2013-14 from A) STC-MODSCAG  retrievals and simulated values from B) WRF-HR and C) WRF-CR runs is shown. Panel D illustrates the distribution of midday mean fSCA over each subregion identified by glacier classification following Randolph Glacier Inventory. The circle and vertical legs represent mean±standard deviation over each region for the entire year. Here, Hindu Kush, Karakoram, W.Himalayas, C.Himalayas, E.Himalayas, Tibetan Plateau, West Kunlun and East Kunlun regions are abbreviated as HK, KK, WH, CH, EH, TP, WKun and EKun, respectively. Panel E shows time-series of daily midday SGS (hashed lines) and fSCA (solid lines) from MODSCAG  (black), WRF-HR (blue) and WRF-CR (red) over a grid located near the Chotta Shingri glacier (marked by magenta diamond in Figure 3A) of western Himalaya region.





Interestingly, the SGS values from WRF-CR, over grids comprising the Chotta Shingri
glacier (marked by magenta diamond in Figure 4A) of western Himalaya, are closer to STC-
MODSCAG observations compared to those from the high resolution model, WRF-HR. As a
sanity check of the above explanation of fSCA-SGS association, daily changes of SGS (hashed
lines in Figure 4E) and fSCA (solid lines in Figure 4E) from STC-MODSCAG (black), WRF-
HR (blue) and WRF-CR (red) over this glacier are compared. Fractional snow cover from STC-
MODSCAG gradually increase (from below 0.2) in November, 13 (to above 0.8) in February
and subsequently decrease back to 0.1 gradually by Septmeber, 14 at the glacier location.
Corresponding SGS values from STC-MODSCAG closely followed the seasonal trend in fSCA
varying around the values of 80-200 micron in winter. In comparison, simulated fSCA from
WRF-HR values drastically increased to 1 in starting of November, 13 (from no snow cover
before that), remain fully snow covered till mid-June, 14 and then steeply become snow free for
rest of period after June. Compared to satellite estimates, fSCA from WRF-HR are greater in
magnitude throughout the duration of snow cover indicating more snow mass simulated by
WRF-HR. Associated SGS values simulated by WRF-HR values are also greater in magnitude
(80-800 micron) than STC-MODSCAG estimates throughout the snow duration over the grid. In
addition, the day to day variation in simulated SGS values is much larger than the STC-
MODSCAG  observations. However, fSCA variation from WRF-CR over this grid is very close
to the variation seen by STC-MODSCAG and the associated SGS values from WRF-CR of (50-
400 micron) are also closer to the estimated STC-MODSCAG values, supporting our argument
of fSCA-linked bias in SGS estimates between model and satellite retrievals.
The annual mean snow albedo (α) values and the distribution over each sub region from
satellite estimates (by combing grain sizes from STC-MODSCAG and decrease in albedo from



STC-MODDRFS (see Bair et al, 2016)) and simulated by both models are presented in Figure 4.
Highest annual mean α values (~ 0.65-0.75) are observed over mountain peaks in Karakoram,
Pamirs and W.Himalaya regions (Figure 5A). The location and magnitude of annual mean α over
these grids are closely reproduced in WRF-HR with an underestimation of < 10% (Figure 5B).
WRF-CR simulated annual mean α values over these grids have a considerably larger
underestimate of ~50% (Figure 5C). Similar statistics are prevalent over all the sub regions of
HMA (Figure 5D). Specifically, the distribution of α values from WRF-HR nearly matches the
observed distribution, but, the distribution of albedos from the coarser model, WRF-CR, are
generally 0.2-0.3 lower when compared to the observations. As above, cloud misidentified as
snow could increase grain sized leading to slightly higher albedos. Also, note the opposite bias
direction for albedo simulated by WRF-HR compared to simulated SGS values. This is intuitive
as smaller snow particles cover greater surface area and therefore reflect more solar radiation
from the surface. A similar pattern in distribution of snow albedo from WRF-HR and WRF-CR
are also found over the sub regions for summer and winter months, separately (Figures 5E and
5F), indicating robust improvement in simulation of albedo values from WRF-HR throughout the
year. The differences in simulated α in WRF-HR with the observations increased during summer
and were lower during winter. Here, it is worth mentioning that we are comparing instantaneous
estimates obtained from Terra overpass during 1000 LT with midday (1000-1400 LT) mean
model values. The inherent diurnally in -α values under clear sky conditions in summer season
(Bair et al., 2017) might contribute partially to the observed enhancement in differences during
summer season. The improvement in α estimation from WRF-HR compared to WRF-CR can be
attributed to the relatively better simulation of the overall macro- and microphysical properties of
the snowpack in high resolution runs.

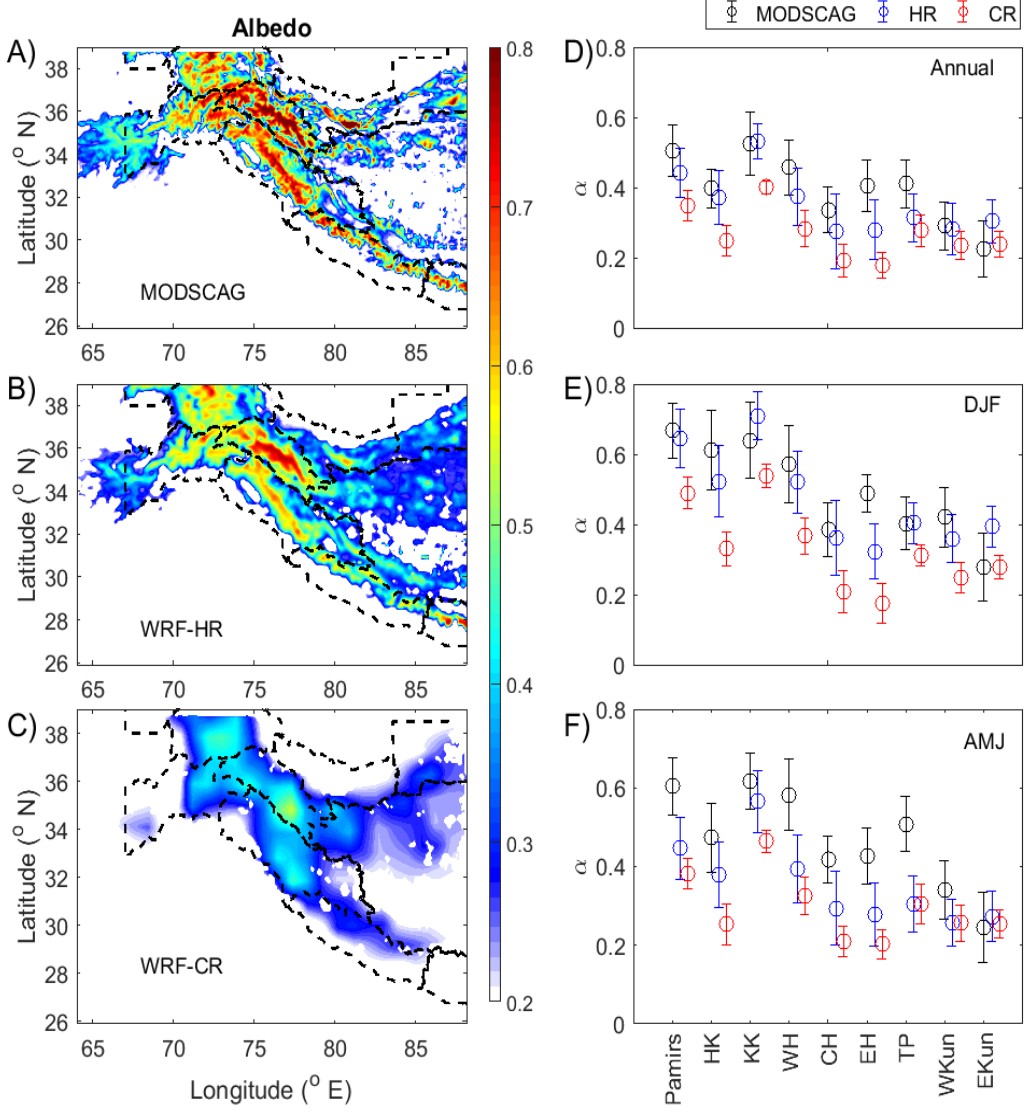

Figure 5: Spatial distribution of annual mean snow cover fraction (fSCA) during midday (1000-
1400 LT) for water year 2013-14 from A) STC-MODSCAG retrievals and simulated values from
B) WRF-HR and C) WRF-CR runs is shown. Panels D-F illustrate the distribution of midday
mean fSCA over each subregion identified by glacier classification following Randolph Glacier
Inventory. The circle and vertical legs represent mean±standard deviation over each region for
D) entire year, E) winter (December - February) and F) summer (April - June) season, separately.
Here, Hindu Kush, Karakoram, W.Himalayas, C.Himalayas, E.Himalayas, Tibetan Plateau, West
Kunlun and East Kunlun regions are abbreviated as HK, KK, WH, CH, EH, TP, WKun and
EKun, respectively.



**3.2: Aerosol distribution and LAP in snow**

We used available in-situ and ground sun photometer measurements from seven different

sites across our study domain (location shown in Figure 6) to evaluate the simulated aerosol

optical depth (AOD). The annual mean midday AOD at each site is shown in Figure 6A. Three

sites (i.e. Merak, CAS and Kathmandu shown in Figures 6B-D) are located on the Himalaya

slopes and the other four sites (Lahore, Jaipur, Kanpur and Gandhi College shown in Figures 6E-

H) are located in the Indo-Gangetic Plains. In-situ measurements clearly illustrate a sharp

decrease (4-5 fold) in mean AOD as we traverse higher up the Himalayan slope. The annual

mean AOD for Lahore and Kanpur sites are 0.41 and 0.52, respectively, while the AOD over

high elevated sites i.e. Merak and CAS sites are 0.07 and 0.05, respectively. Also, MODIS-

observed AOD values prominently show the reduction in annual mean AOD from the Indo-

Gangetic Plains (MODIS-AOD ~ 0.4-0.7) to the Tibet region (MODIS-AOD ~ 0.1-0.2) (Figure

S5). Over the four sites in the Indo-Gangetic Plains, AOD simulated by both WRF-HR and

WRF-CR runs are well correlated with observations (r= 0.5-0.6, Figures 6E-6H). The biases in

modelled AOD are also similar (in the range of 0.2-0.4) in case of both WRF-HR and WRF-CR

runs (Figures 6E-6H). Thus, no significant improvement in AOD values are achieved over the

plain region with fine resolution. However, distinct and large improvement in simulated AOD is

seen over the high elevation sites due to the increase in spatial resolution. Note that AOD values

from WRF-CR are not strongly correlated with observations at these sites (Figures 6B-6D) and

also have very high positive biases in AOD values (even higher than annual mean values at

Merak and CAS stations). In contrast, the correlation between observations and WRF-HR is

reasonably good (r=0.5-0.8 at these sites) using fine spatial resolution in WRF-HR. The positive

biases in AOD from WRF-HR at Merak and CAS sites are lower than corresponding WRF-CR



values by an order of magnitude. Presence of lower biases in AOD from WRF-HR over high
elevation sites indicates that the observed sharp decrease in AOD values across the Himalayan
slope are better captured by the higher resolution WRF run (WRF-HR) than in the coarser run.
Greater annual mean AOD value is simulated by WRF-CR over the entire HMA region
compared to WRF-HR (Figure S5) supporting an overestimation of AOD from WRF-CR at
higher elevation in addition to the few sites.  The presence of high biases (0.3-0.4) over
Kathmandu valley even in WRF-HR runs indicate that model resolution even finer than 12 km is
likely needed to better resolve the AOD distribution in complex terrain around valleys in
Himalayan slope regions. Moreover, Jayarathne et al., 2018 shows that many local emissions are
not accounted in global emissions which causes underestimation in simulated regional AOD
values in these valleys. Temporal variability in monthly mean AOD (relatively higher AOD in
summer) is simulated reasonably well by both the model versions (Figure S5).











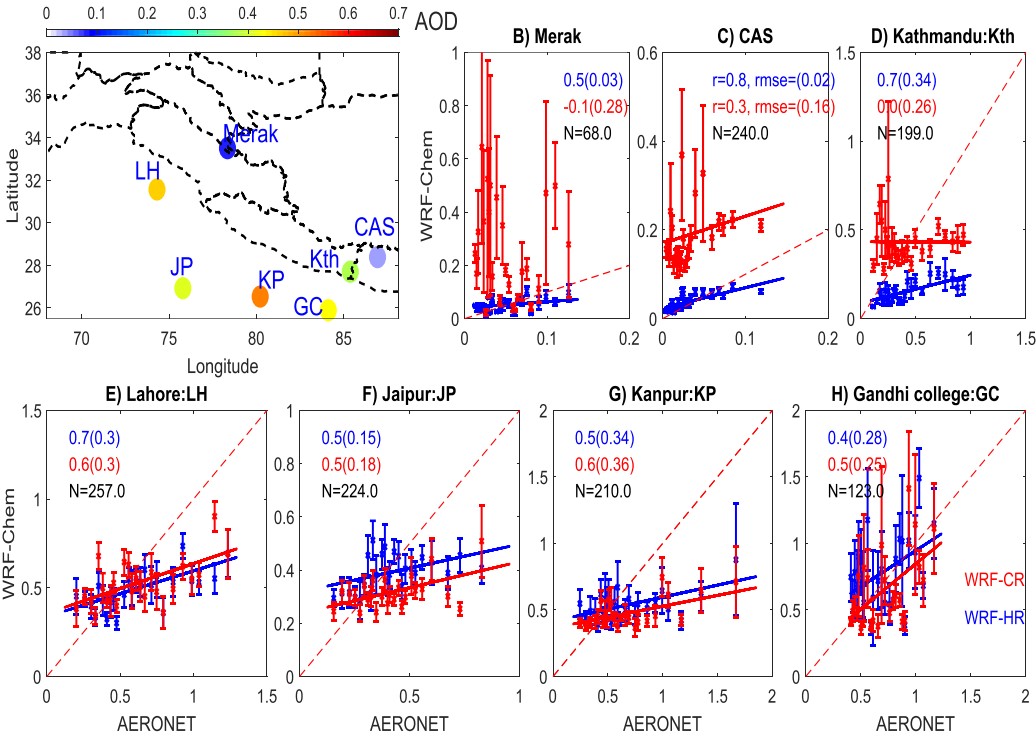


Figure 6: Comparison of midday (averaged over 1000-1400 LT) aerosol optical depth (AOD)
measured by ground based sun photometer at seven sites within the study domain with
corresponding simulated AOD values from, both, WRF-HR and WRF-CR. The annual mean
AOD values over each site is shown by shade in topmost left Panel. The other panels illustrates
the comparison over one of each stations shown by dots in topmost left Panel. The 'N'
mentioned in the legend in each panel is the total number of days when collocated data between
model and measurement is available over that site. These sample points are divided into 50 equal
bins of ascending AERONET-AOD values (2 percentile each) and averaged. The standard
deviation in each bin is shown by the vertical bars. The correlation coefficient values (r) are also
mentioned in the legend followed in brackets by the relative error values ($\sum$rmse/mean obs).








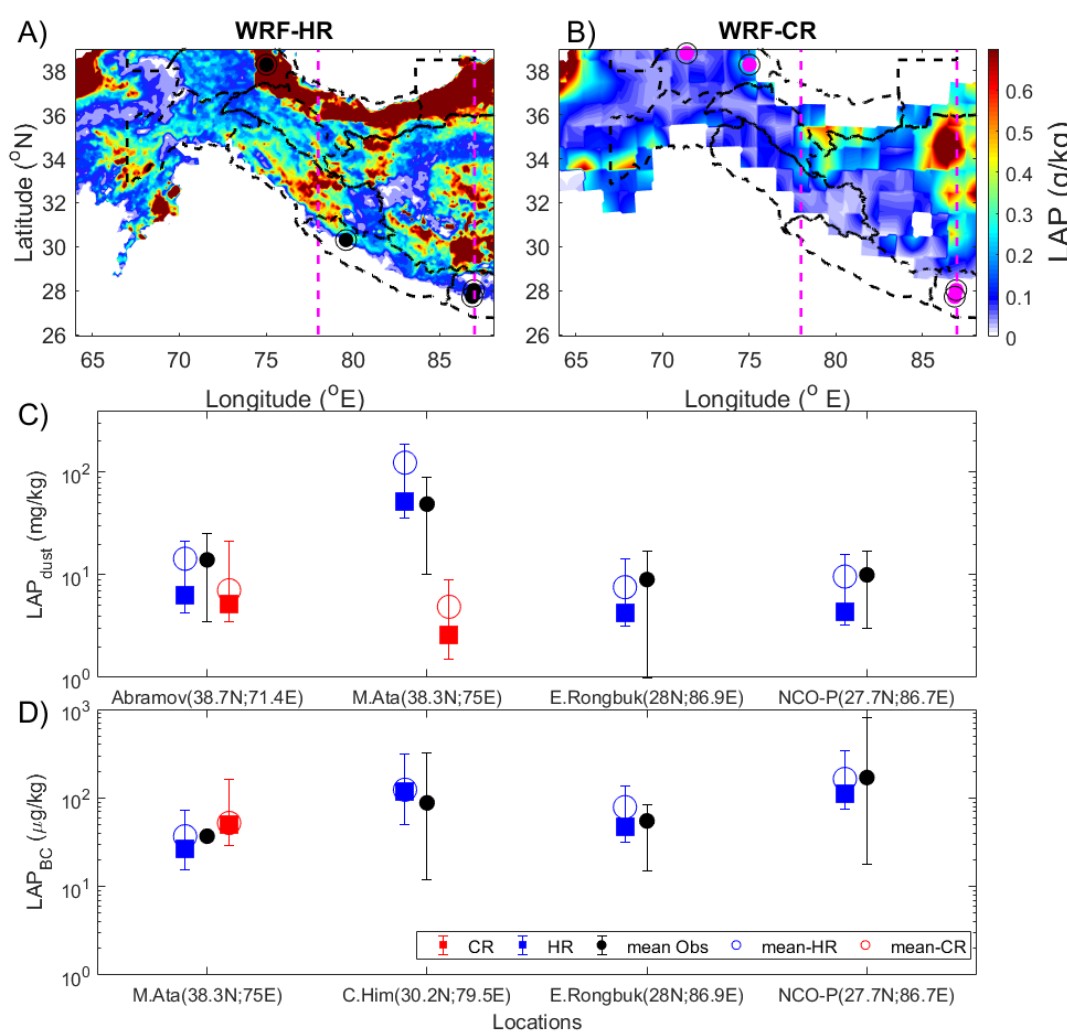


Figure 7: Spatial distribution of annual mean LAP concentration in snow top layer for water year
2013-14 simulated by A) WRF-HR and B) WRF-CR. The black dots in Panel A denote the
locations where observations of BC in snow is available. Similarly, the magenta dots in Panel B
denote the locations from where observation of dust in snow is available. Panels C-D illustrate
comparison of simulated $LAP_{BC}$ (top) and $LAP_{dust}$ (bottom),respectively, in topmost snow layer
with observed values over the marked locations in the Himalayan cryosphere. Annual mean
(circle) and distribution (box plot) is used as metric of comparison. The WRF-HR and WRF-CR
values are represented by blue and red, respectively. The pink lines in Panel A are the cross-
sections shown in Figure 8.



Significant differences in simulated AOD over high elevations of Himalaya slopes and
TP indicate that considerable differences might also be present in LAP concentrations in snow
between the two WRF simulations. Annual mean LAP concentrations in top snow layer from
WRF-HR and WRF-CR simulations are compared in Figures 7A-7B. The comparison shows that
LAP concentration in WRF-HR are significantly higher than WRF-CR simulated values.
Quantitatively, the WRF-HR simulated annual mean LAP concentrations over the Pamir (0.5
g/kg), Karakoram (0.45 g/kg), Hindu Kush (0.2 g/kg), W. Himalaya (0.3 g/kg), C. Himalaya (0.2
g/kg) and E. Himalaya (0.08 g/kg) ranges is 3-5 times higher than the same from WRF-CR runs.
In contrast, WRF-HR simulated LAP over TP (0.21 g/kg) and Kunlun ranges (0.8 g/kg) is similar
to the mean magnitude simulated by WRF-CR runs. As a sanity check, we evaluate the simulated
LAP concentrations against those previously reported in the literature. Figures 7C and 7D
illustrate the evaluation of mean annual LAP concentration from WRF-HR and WRF-CR
associated with BC ($LAP_{BC}$) and dust ($LAP_{dust}$), respectively, against the reported data (shown as
filled black circles). The locations of reported $LAP_{BC}$ (black filled dots) and $LAP_{dust}$ (magenta
filled dots) are shown in Figure 7C and Figure 7D, respectively. First, the difference in the
magnitude of $LAP_{dust}$ and $LAP_{BC}$ over HMA is striking. The $LAP_{dust}$ is more than 1000 times
greater than $LAP_{BC}$ both in observations and the models. Secondly, $LAP_{BC}$ and $LAP_{dust}$ values
from WRF-HR are much closer to reported values compared to WRF-CR values. The differences
in mean of reported $LAP_{BC}$ and $LAP_{dust}$ distribution to that simulated by WRF-HR at various
sites are in range of 5-30 μg/kg and 5-20 mg/kg, respectively. WRF-CR well simulates the
concentrations of $LAP_{BC}$ and $LAP_{dust}$ over Pamirs (~ 10 mg/kg), but significantly underestimates
the $LAP_{BC}$ and $LAP_{dust}$ (by an order of magnitude) over the Himalayan ranges. Although the
reported data are not specific to water year 2013-14, it can be reasonably assumed that the inter-



annual variations of LAP concentration in snow is of the order of magnitude as uncertainty in the
observations. Thus, the WRF-HR better simulates aerosol and LAP concentration than the WRF-
CR over the HMA region.

It is interesting to note that finer spatial resolution resulted in lower AOD but greater

LAP values in snow over some places in HMA.  For more insight, the vertical distribution of
aerosol concentration in altitude-latitude space (Figure 8) across two latitudinal cross-sections
(magenta colored lines in Figure 7A) is analyzed. Figure 8 illustrates the differences in simulated
vertical distribution of mean aerosol number concentration along 78ºE (row 1) and 87ºE (row 2)
for WRF-HR (left column) and WRF-CR (right column) runs, respectively. Corresponding
terrain elevation (black solid line) and snow depth (magenta bars) are also overlaid in these plots.
The latitude-altitude plots clearly illustrate that improved representation of the terrain in WRF-
HR shows the sharp change of elevation over Himalayan foothills and causes a steeper natural
barrier to the transport of aerosols uphill from IGP to HMA region than in the WFR-CR model.
Also, high spatial resolution enhances snowfall in WRF-HR over the HMA region relative to the
WRF-CR model. While the former change increased annual dry deposition flux, more snowfall
caused greater wet deposition annually in WRF-HR compared to WRF-CR (Figure S6).  The
combination of these effects increases the deposition of aerosols and therefore LAP on the
southern slopes of Himalaya in the WRF-HR run. This explains the coexistence of higher LAP
concentration/deposition and lower AOD across HMA in WRF-HR, compared to corresponding
WRF-CR results.





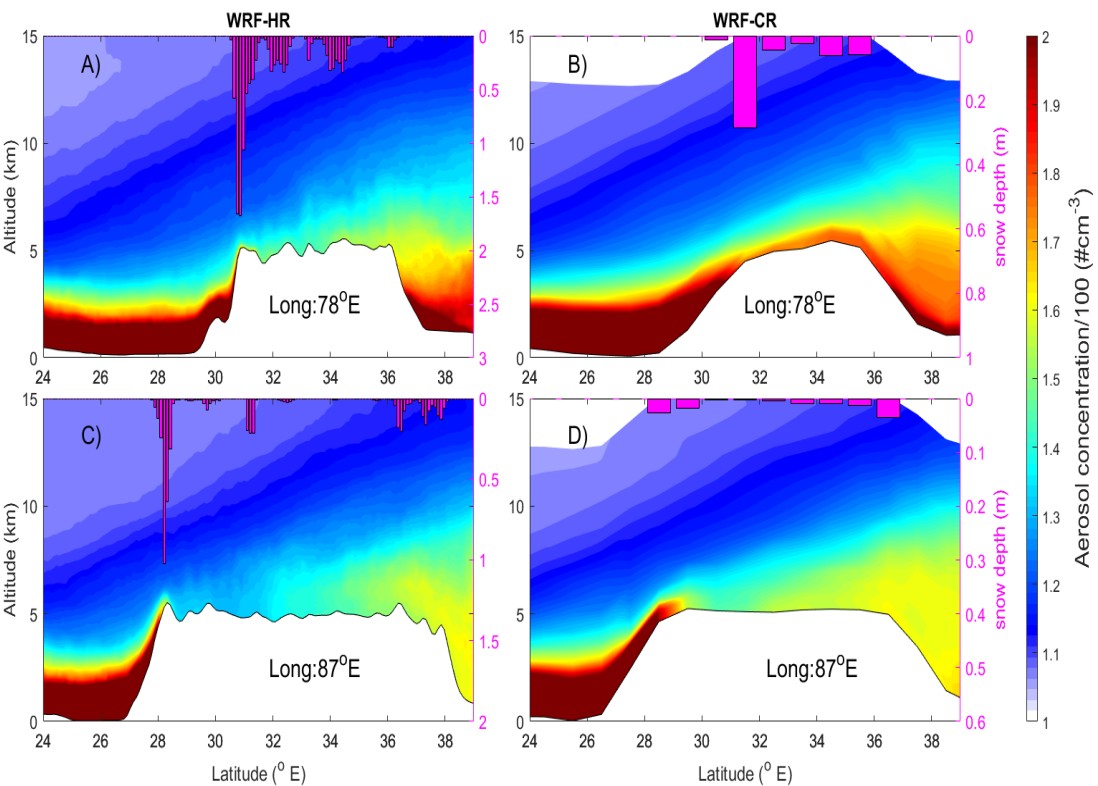

Figure 8: Longitudinally-averaged annual mean aerosol number concentration plotted in altitude-
latitude space for two longitudinal traverses across the study domain, i.e. 78ºN (Panels A and B)
and 87ºN (Panels C and D) for both WRF-HR (left column) and WRF-CR (right column).
Corresponding terrain elevation is shown in solid black line. Corresponding to each latitude, the
longitudinally-averaged annual mean snow depth is also presented in magenta color bars (using
y-axis on the right).







**3.3 LAP-induced Snow darkening and radiative forcing**

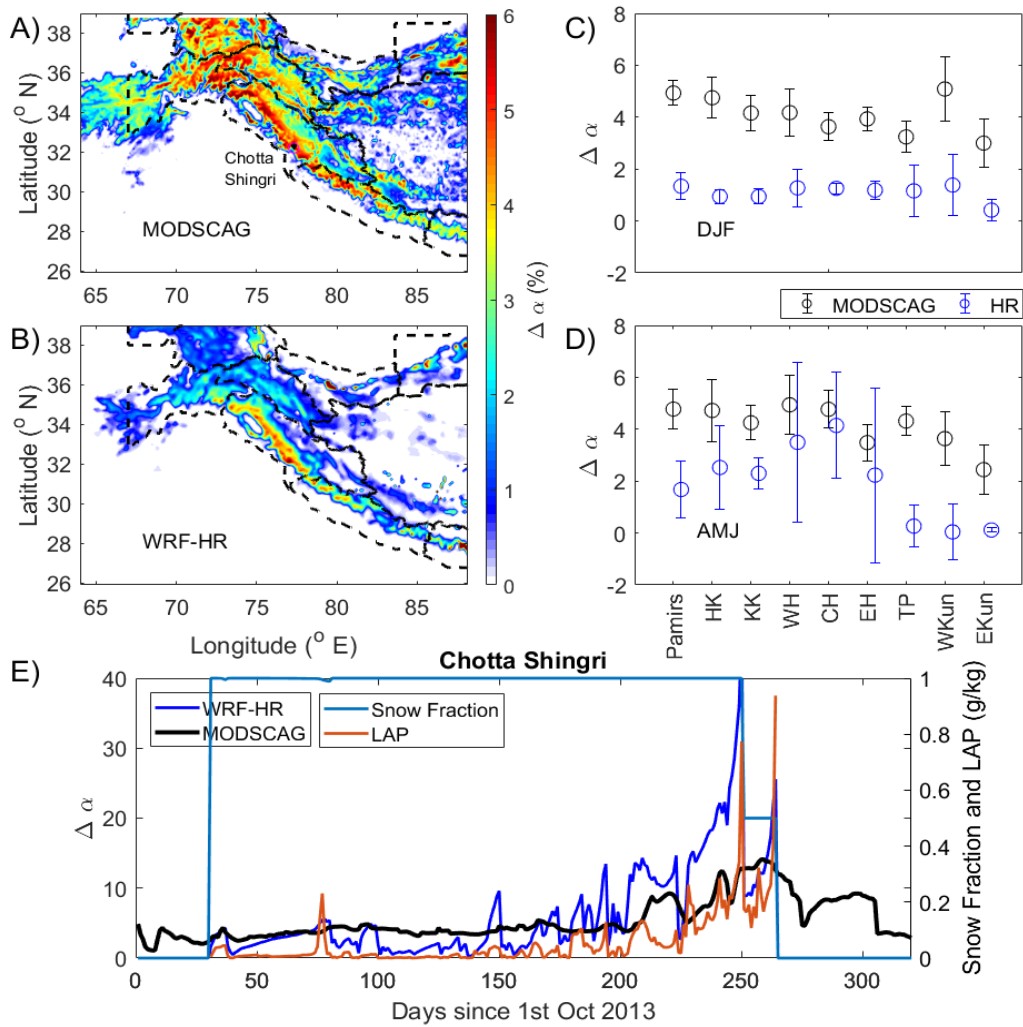

Figure 9: Spatial distribution of annual mean LAP-induced snow albedo darkening (Δα) during
midday (1000-1400 LT) for water year 2013-14 from A) MODSCAG retrievals and B) simulated
values from WRF-HR is shown. Panels C and D illustrate the distribution of midday mean Δα
over each sub region identified by glacier classification following Randolph Glacier Inventory.
The circle and vertical legs represent mean±standard deviation over each region for entire year.
Here, Hindu Kush, Karakoram, W.Himalayas, C.Himalayas, E.Himalayas, Tibetan Plateau, West
Kunlun and East Kunlun regions are abbreviated as HK, KK, WH, CH, EH, TP, WKun and
EKun, respectively. Panel E shows time-series of daily midday Δα from STC-MODSCAG
(black) and WRF-HR (blue) over the grids located near the Chotta Shingri glacier (marked in
Figure A) of western Himalaya region. Also fractional snow cover and LAP concentrations from
WRF-HR over the same grids are included.




STC-MODDRFS retrievals illustrate that locations in Hindu Kush and W. Himalayas
have the highest annual mean LAP-induced reduction in snow albedo ($\Delta\alpha$ in %) followed by
Karakoram,  C.Himalayas and Pamir regions (Figure 9A). WRF-HR simulated the spatial
variations in annual mean $\Delta\alpha$ reasonably well (Figure 9B), but, the magnitudes are
underestimated by ~20-40 % throughout the domain. Note that the biases in annual mean values
are lowest over grids in Himalayan ranges (where the underestimation is within 20%). Season
wise and region wise distribution plots show that the WRF-HR biases are higher in winter
months than the summer months (Figures 9C and 9D). While, WRF-HR simulated $\Delta\alpha$ values in
the winter span between 1-3 %, the corresponding STC-MODDRFS estimates of $\Delta\alpha$ are larger
with values ranging between 3-6% (thus no overlap with model values) over all the sub regions.
In summer months, the distribution of modeled WRF-HR $\Delta\alpha$ values over Karakoram and
Himalayan ranges are similar in magnitude, explaining the lower biases in annual mean values
over Himalayas. This spatiotemporal variability in differences between STC-MODDRFS
retrievals and simulated $\Delta\alpha$ values is consistent with the variability in biases of fractional snow
cover seen in WRF-HR (Figures2E and 2F). Specifically, the large underestimation [and
significant improvement relative to WRF-CR] in WRF-HR-$\Delta\alpha$ values (Figures 9C and 9D) in
winter [summer] over Karakoram, Hindu Kush and Himalayas is in agreement with
corresponding overestimation [improvement] of fSCA from WRF-HR over these regions (Figure

2).

For a closer look, the difference in daily midday mean $\Delta\alpha$ values from STC-MODDRFS
(black) and WRF-HR (blue) are compared (in Figure 9E) over the grids of Chotta Shingri glacier
(similar as Figure 4E). Corresponding, midday mean fSCA (light blue) and LAP (orange) from



WRF-HR are also plotted. Δα values from STC-MODDRFS are about 5% during winter months,
but, increases in summer months until mid-June (peak value is 18%). Albedo reduction is closely
associated with the temporal progression in midday LAP concentration in snow over this region
at daily scale. In agreement, midday Δα values from WRF-HR are lower in winter months and
higher in summer months. Except occasional peaks, with magnitudes of 3-4 %, Δα values from
WRF-HR largely remained below 3 % till late February. A steep increase in Δα values from
WRF-HR is seen in March (monthly mean ~ 4%), April (9%), May (13%) and June (18%). As
already discussed, the simulated fSCA values in WRF-HR are greater than observed fSCA from
STC-MODSCAG for most of the winter season (Figure4E). STC-MODDRFS estimated Δα is
based on surface reflectance, while Δα calculated by model is for a surface layer of ~3 cm. The
surface snow layer in SNICAR/CLM continuously evolves as fresh snowfall is added or with
snow melting, so the LAP concentrations in surface layer depend on new snowfall, meltwater
flushing, and layer combination/division (Flanner et al., 2007; Flanner et al., 2012; Oleson et al.,
2010). Thus, more snow cover or thicker surface layers in winter results in lower values of
annual mean LAP concentration and thus underestimates associated with LAP-induced snow
darkening. In addition, the associated overestimation in modelled SGS during winter (Figure 4E)
can also contribute to the lower WRF-HR-Δα values, because, bigger snow grains in WRF-HR
lead to lower clean albedo and thus smaller reduction in albedo compared to STC-MODDRFS .

Another notable point is the large Δα values (> 20 %) from WRF-HR that occur towards

the end of snow cover in June, which, is not seen in the STC-MODDRFS retrievals. The
variations could be due to 2 reasons, either the snowpack is underestimated or the LAP
concentration is overestimated by the model. Some factors which can contribute to these
discrepancies in summer are 1) Larger LAP values may be simulated due to model uncertainties



in enhanced wet scavenging fluxes; 2) It is well known that with transport and deposition of
aerosols from IGP to W.Himalayas increases afternoon with evolution of boundary layer over the
IGP region (Dumka et al., 2015;Raatikainen et al., 2014). This feature is well simulated by the
model (not shown). As STC-MODDRFS estimates are representative of 1000 LT, but, modelled
values are representative of midday mean (1000-1400 LT), more aerosol deposition might be
resulting in higher $\Delta\alpha$ values WRF-HR during summer months; 3) At the same time, the
uncertainties associated with modelling aerosol-snow albedo microphysical feedbacks to snow
melt may also be contributing to underestimation of snow packs in summer. However, more
certainty on these modelled values require evaluation against in-situ measurements. It is worth
mentioning here that no in-situ measurements are available for direct comparison of these high
$\Delta\alpha$ values WRF-HR over W.Himalayas (Gertler et al., 2016). Nonetheless, the high values
simulated during summer end are in near-range of previously reported values over other HMA
regions. Kaspari et al., (2014) used the offline SNICAR model to report that BC concentrations
in spring snow/ice samples at Mera Glacier were large enough to reduce albedo by 6-10% but
that with the inclusion of dust, the reduction in albedo was 40-42% relative to clean snow.
Recently, Zhang et al., 2018 has combined a large dataset of LAP measurements in surface snow
with offline SNICAR model to illustrate that $\Delta\alpha$ can be >35% over Tibetan Plateau. Moreover,
the composite effect of this discrepancy on seasonal/annual mean values is minimal as the
snowpack is at its minimum at summer end. Similar high daily variability and huge LAPRF
values (~200 W/m$^2$) in late summer as well as associated sudden decline in snow depth is also
reported in sites over upper Colorado river basin (Skiles et al., 2015; Skiles and Painter, 2017).





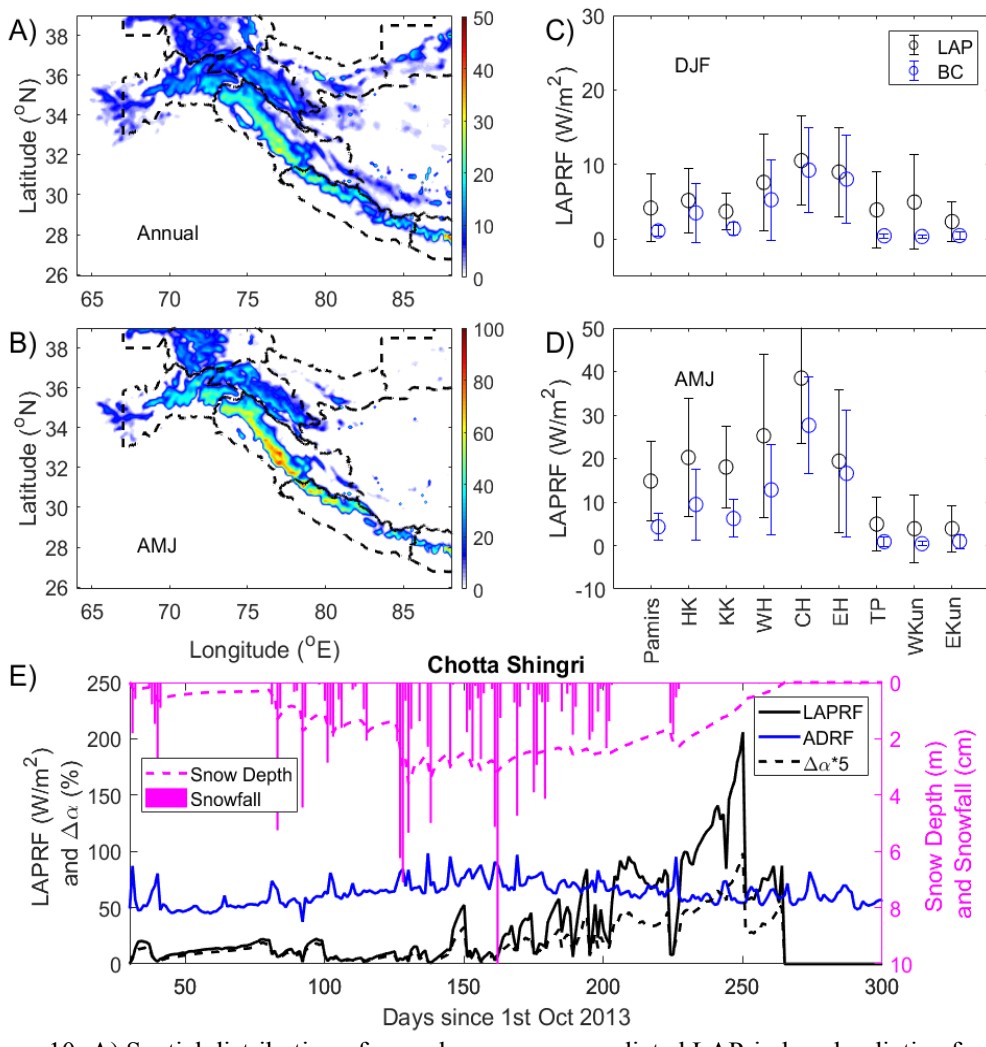

Figure 10: A) Spatial distribution of annual mean snow mediated LAP-induced radiative forcing
(LAPRF) from WRF-HR and B) Spatial distribution of seasonal mean LAPRF over snow
covered for summer season (AMJ) of water year 2013-14. Panels C and D illustrate the
distribution of LAPRF (blue) and BC-only-LAPRF (black) over each sub region identified by
glacier classification following Randolph Glacier Inventory for winter and summer, respectively.
The circle and vertical legs represent mean±standard deviation over each region for entire year.
Here, Hindu Kush, Karakoram, W.Himalayas, C.Himalayas, E.Himalayas, Tibetan Plateau, West
Kunlun and East Kunlun regions are abbreviated as HK, KK, WH, CH, EH, TP, W.Kun and
E.Kun, respectively. Panel E shows time-series of daily midday LAPRF (blue) and aerosol direct
radiative forcing at surface (ADRF, blue) over the Chotta Shingri region (marked in Figure 9A)
of western Himalaya region. The simulated LAP-induced change in snow albedo is also shown



in hashed black line. Also snow depth and snowfall from WRF-HR over the same grid is
included.
In agreement with the spatial variation in Δα values, the annual mean snow-mediated
LAP-induced radiative forcing (LAPRF) from WRF-HR over grids in W.Himalayas and C.
Himalayas are highest followed by Karakoram, Hindu Kush, E.Himalayas and Pamir regions
(Figure 10A). The spatial distribution of summer mean LAPRF values from WRF-HR (Figure
10B) is similar to that of the annual mean LAPRF values, but, the summer magnitudes are higher
by an order of magnitude throughout most of the domain. This is mainly due to the large increase
in Δα values in summer when LAPs aggregate on the surface compared with winter months
when LAPs are continuously covered by new snow (Figure 9). Spatio-temporal variability in
LAPRF is evident in the seasonal and regional distribution plots (Figure 10C and 10D). LAPRF
values over the edges of HMA are greater than the highland TP region in both winter and
summer months probably due to greater LAP deposition simulated over Hindu Kush, Karakoram
and Himalayas regions (Figure S7) from the close proximity of dust sources. Also, it is clearly
visible that the maximum LAPRF values within HMA region are present over grids in
Himalayan ranges (during both winter and summer) with annual mean values $> 50$ W/m$^2$ (seen in
Figures10B and 10D) and maximum instantaneous values higher than 150 W/m$^2$ (not shown).
The time series of midday mean LAPRF values (Figure 10E) over the same grids in the Chotta
Shingri Glacier region in western Himalaya is plotted to ascertain possible daily variability in
LAPRF for midday over the region. Corresponding LAP-induced albedo reduction, snow depth
and snowfall values are also plotted to show how LAPRF can affect local snow melting. The
daily snow depth increases to 3.6 m in winter (30-150 days) followed by gradual reduction and
snow cover melting in summer (150-270 days). The mean midday LAPRF value is ~10 W/m$^2$ in
winter season, but, the magnitude increased gradually during March (18 W/m$^2$), April (44



W/m$^2$), May (86 W/m$^2$) and June (123.5 W/m$^2$), eventually terminating with a peak value of 202
W/m$^2$ in mid-June. The temporal evolution in $\Delta\alpha$ values closely followed the LAPRF values with
a variation in range of 1-20%. The shortwave aerosol direct radiative forcing (ADRF) during
midday at the surface over the same grid is also shown (as blue curve) in Figure10E. The
momentarily high values of ADRF during the period indicate dust storms or sudden increase in
aerosol loading over the grid. A closer look illustrate that the large daily variations in LAPRF
and $\Delta\alpha$ are associated with the variability in ADRF and daily snowfall occurence over the grid.
Fresh snowfall feedbacks result in subsequent reduction in LAPRF and enhancement in snow
depth (Figure10E), while, higher aerosol loading over aged snow is followed by a clear increase
in LAPRF. During melting, our model considers that only a fraction of LAP is washed away with
meltwater, which, results in enhanced concentration of LAP during the end stages of the
snowpack. This snow albedo feedback along with the momentary high aerosol loading (on 250$^{th}$
day) can explain the very high values of albedo reduction and LAPRF that were simulated during
the last days of the snow cover over this grid. Higher LAPRF values indicate more energy being
absorbed by the snow pack and thus more snow melting. Therefore, LAP-induced snow melting
effect over Himalaya is very significant and is the largest relative to other areas within HMA.

Snow-mediated radiative forcing only due to BC- deposition is shown in Figures10C and

10D. Although, similar spatiotemporal pattern in LAPRF and BC-only-LAPRF is simulated,
contrasting features in context to BC contribution to LAPRF are present in between the western
part and eastern parts of HMA. During winter, the magnitude of BC-only LAPRF values is
similar to that of total LAPRF over the western part of HMA region (i.e. Pamirs, Karakoram,
Hindu Kush and W.Himalayas) suggesting that BC is a dominant contributor to net LAPRF
(Figures10C). But, large differences between LAPRF and BC-only LAPRF is present during



summer season over the western regions indicating that dust contribution is more or less equal to
that of BC contribution over these regions (Figure10D). In contrast, the dust contribution to
LAPRF values over the eastern domain (TP and Kunlun regions) is significant during winter
season (Figures10C and 10D). The spatiotemporal variability in dust and BC contribution is
reported previously and is mainly linked with the seasonal variability in meteorology and
associated advection of South Asian emissions (Zhang et al., 2015; Wang et al., 2015a ; Niu et
al., 2018). The ADRF-induced surface cooling effect may nullify the effects of LAPRF-induced
warming effect on snowpack melting. But, the drastic increase in LAPRF values in April through
June causes the magnitude of LAPRF to be twice that of ADRF over W. Himalayas during snow
melting period, highlighting dominance of LAPRF (as also seen in Figure10E).
The simulated annual mean, summer mean and BC-only-LAPRF values from WRF-HR
are in general, higher compared to previously reported estimates of LAPRF from model studies
at coarser resolution. For example, Ménégoz et al., (2014) have reported annual mean LAPRF of
~ 1-3 W/m$^2$ over Himalaya using an online simulation at 50 km resolution grid. Similarly, Qian
et al., (2011) used the Community Atmosphere Model version 3.1 at coarser spatial resolution to
show that simulated aerosol-induced snow albedo perturbations can generate LAPRF values of
5-25 W/m$^2$ during spring over HMA. Also, coarsely resolved GEOS-Chem runs simulated BC-
only-LAPRF can vary from 5 to 15 W/m$^2$ in the snow-covered regions over the TP (Kopacz et
al., 2011).  Recently, a decade long simulation using the RegCM model at 50 km spatial
resolution also estimated maximum BC-only-LAPRF values of 5-6 W/m$^2$ over the Himalaya and
southeastern TP averaged over non-monsoon season (Ji, 2016). However, the comparison of
WRF-HR and WRF-CR simulations provided in this study clearly show that the magnitudes of
snow macro- and micro-properties, aerosol loading and LAP-induced albedo darkening over



Himalayas improved significantly with finer spatial resolution. Thus, the global model simulated
LAPRF values are likely underestimated. In agreement, recently, Zhang et al., (2018) and has
estimated BC-only-LAPRF of 20-35 W/m$^2$ using offline SNICAR calculation forced with a
greater coverage of measurements of surface snow content. He et al., (2018) have also reported
similar high BC-only-LAPRF values after implementing a realistic snow grainsize
parameterizations in offline SNICAR calculations over HMA.
**4. Summary and Implications**
In this study, we use the SNICAR model coupled with WRF-Chem regional model at high
spatial resolution (WRF-HR; 12 km) to simulate the transport, deposition, and radiative forcing
of light absorbing particles (LAPs; mainly Black Carbon and dust) over the high mountains of
Asia (HMA) during water year 2013-14. The snow grain sizes and LAP-induced snow darkening
was evaluated, for the first time, against comprehensive satellite retrievals (the MODSCAG and
MODDRFS spatial and temporally complete retrieved satellite observations) over HMA. The
atmospheric aerosol loading is evaluated against satellite and ground-based AOD measurements
over HMA region. Results from another simulation which employ the same model configuration
but a coarser spatial resolution (WRF-CR; 1 degree) are also compared with WRF-HR to
illustrate the significance of a better representation of terrain on snow-pack and aerosol
simulation over HMA. The main conclusions from our study are:

856a)    The simulated macro- and micro-physical properties and the duration of snow packs over HMA

improve significantly due to the use of fine spatial resolution, especially over the southern slopes
of Hindu Kush and Himalayan ranges.

859b)    Simulated aerosol loading over HMA is also more realistic in WRF-HR than in WRF-CR, which

leads to a reduction in biases of annual mean LAP concentration in snow. This improvement is



attributed to a more realistic simulation of wet deposition (due to a better simulation of snow
pack) and dry deposition of LAPs (associated with a better representation of terrain) in WRF-
HR.
c) WRF-HR captures the magnitude of LAP-induced snow albedo reduction ($\Delta\alpha$) over Himalayas
and Hindu Kush region relatively well compared to the STC-MODDRFS retrievals during
summer. However, during winter, large biases in modelled $\Delta\alpha$ values are present. This is
probably due to inherent uncertainties in model parameterizations and satellite retrievals
associated with the cloud cover over HMA in winter period.
d) The glaciers and snow cover regions located in the Himalaya have the highest LAPRF within
HMA i.e. annual mean LAPRF $\sim$ 20 W/m$^2$ and summer mean LAPRF $\sim$ 40 W/m$^2$. This is
consistent with similar high values of $\Delta\alpha$ over Himalayan ranges i.e. annual mean $\Delta\alpha$ values $\sim$ 2-
4 % and summer mean $\Delta\alpha$ values $\sim$ 4-8 %. The annual mean LAP concentration in snow values
(200-300 mg/kg) are also high. Thus, the Himalaya (more specifically, western Himalayas) is
most vulnerable to LAP-induced snow melting within HMA.

Ramanathan and G. Carmichael, (2008) suggest that atmospheric warming from LAPs
($\sim$20 W/m$^2$) may be just as important as greenhouse gases in the melting of snowpack and
glaciers over Asia. In this context, the high magnitudes of LAPRF values in summer over HMA
($\sim$ 40 W/m$^2$) clearly shows that snow-mediated aerosol forcing on snow melting over HMA can
be twofold the significance of the atmospheric forcing. Nonetheless, the differences in snow
surface properties between WRF-HR and satellite observations indicate probable uncertainties in
model parameterizations. At the same time, the STC-MODDRFS retrievals themselves may have
an uncertainty of $\sim$ 5% in instantaneous $\Delta\alpha$ measurements. Thus, efforts to further reducing the



LAPRF uncertainties in the model are warranted in the future by using in-situ observations (i.e.
field campaigns), specifically over the most affected western Himalayas, where relevant
measurements are largely absent (Gertler et al., 2016). Moreover, satellite retrievals will be
markedly improved in the coming decade with the NASA Decadal Survey Surface Biology and
Geology imaging spectrometer mission, which includes as a core measurement snow albedo and
its controls (National Academies of Science, 2018).  These visible through shortwave infrared
imaging spectrometer retrievals have uncertainties an order of magnitude smaller (Painter et al.,
2013) than those from multispectral sensors such as MODIS and will provide a more accurate
constraint on the physically-based modeling pursued here.

**Acknowledgment**
This research was supported by the NASA High Mountain Asia Project. The Pacific Northwest
National Laboratory (PNNL) is operated for DOE by Battelle Memorial Institute under contract
DE-AC06-76RLO 1830.  Part of this work was performed at the Jet Propulsion Laboratory,
California Institute of Technology under contract with NASA. H. Wang acknowledges support
from the U.S. Department of Energy (DOE), Office of Science, Biological and Environmental
Research as part of the Earth System Modeling program. We thank the PIs of the selected
AERONET and SKYNET stations, for providing the data used in this study. The AERONET
data are obtained from the AERONET website, http://aeronet.gsfc.nasa.gov/. We also thank the
PI of the selected SkyNet station, for providing the data used in this study. The SkyNet data is
obtained from the website http://atmos3.cr.chiba-u.jp/skynet/merak/merak.html/.



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
