# Peer review of "Impact of light-absorbing particles on snow albedo darkening and associated radiative forcing over High Mountain Asia: High resolution WRF-Chem modeling and new satellite observations"

_Atmospheric Chemistry and Physics, 2018_

## Referee Comment (RC1) · Anonymous Referee #1 · 11 Feb 2019

Based on model simulations and recent satellite observations, the authors examine the skill of high resolution WRF on the impact of snow albedo darkening on radiative forcing. They evaluate the model simulation with various obsevations. The authors also discussed the benefit of high resolution model, by comparing with coarser resolution model simulation, in aerosol and snowpack distribution over HMA. Lastly, spatial/temporal variation of radiative response are also discussed. This is well organized and clearly written manuscript. I would like to suggest this work for publication subject

to a revision to address following concerns.

1) Snow albedo: How was snow albedo estimated when snow cover fraction is less than 1? For example, is albedo difference between WRF-HR and WRF-CR (Fig. 5) due to snow cover fraction or aerosol concentration (LAP)? If snow cover fraction is an important factor, can model snow albedo be compared with STC-MODDRFS?

2) Season: Winter and summer are defined as DJF and AMJ in main text while NDJF and MAMJ are used in supplementary material. Spring is more appropriate for AMJ as well as MAMJ.

3) Line 336: 16:00LT → 14:00LT

4) Validation of LAP (Lines362-370): What would be the effect of using LAP data observed in different year and different glaciers?

5) Lines 416-418: It seems obvious, but any thought why model overestimate snow cover fraction in DJF and underestimate in AMJ? Does model simulate reasonable surface temperature and precipitation over the region?

6) Line 461: Is "and 4F" typo?

7) Lines 498-503: Hard to read. Are "13" and "14" day of the month?

8) Lines 509-512: It's hard to follow the argument. Would you get the same conclusion if a different grid is used?

9) Line 515: Fig. 4 → Fig. 5

10) Fig. 4 has a wrong caption

11) Fig. 5 has a wrong caption

12) Fig. 7A: What cause the high LAP in the south of Taklamakan desert?

13) Spatial pattern of LAP(Fig. 7) doesn't seem to highly correlated with albedo (Fig. 5) distribution.

14) AOD is higher in CR, but LAP is higher in HR. Yet, albedo is higher in HR in general. It is probably due to higher snow cover (See also my comment #1) and more snowfall (Lines 654-657) in HR. If so, additional simulation (with similar climatology) may be required to discuss the impact of snow darkening.

15) Fig. 9C & Lines 866-868: What cause the bias? Does model has local source of dust over HMA?

16) Same emission databases on relatively coarse resolution are used for both simulations. How about actual emission of dust and BC? Are they comparable in HR and CR?

---

## Referee Comment (RC2) · Anonymous Referee #3 · 21 Mar 2019

This study performed WRF-chem high-resolution (∼12km) model simulations coupled with the SNICAR model to resolve BC/dust-snow interactions and associated albedo reductions. The model results are evaluated with a suite of observations for snow properties and aerosol conditions. The authors further illustrated the effect of model resolution by comparing with a quasi-global WRF-Chem simulation. This study highlights the importance of model resolution and aerosol-snow interactions in modeling the Tibetan Plateau hydroclimate. The manuscript is generally well-organized and well-written. I

have a few comments and suggestions to help improve the manuscript.

1. When evaluating modelled LAP concentration in snow, did the authors average the LAP concentrations throughout all the snow layers (up to 5) in the model? Or did the authors only look at specific snow thickness (e.g., top 10 cm) in the model? The observed LAP in surface snow may have different sampling thicknesses/depths. More clarifications could be useful. Similar clarifications should also be provided for snow grain size calculations and evaluation.

2. Introduction (Lines 112-114): For the authors' information, there are a number of valuable recent global modeling studies on the LAP-induced snow albedo effect over the Tibetan Plateau which could be included here as references (e.g., Kopacz et al. 2011 ACP, https://doi.org/10.5194/acp-11-2837-2011; He et al. 2014 GRL, https://doi.org/10.1002/2014GL062191; Zhang et al. 2015 ACP, https://doi.org/10.5194/acp-15-6205-2015).

3. Line 343: It is a little weird to identify April-June as the summer season. How about using "pre-monsoon" season?

4. Is there any specific reason to select 2013-2014 water year as the simulation period? Is it because the availability of observations?

5. Another important uncertainty factor the authors did not mention is the snow grain shape effect. Recent studies have shown significant effects from non-spherical snow grain shapes on snow albedo and BC/dust-induced albedo reduction (e.g., Liou et al. 2014 JGR, https://doi.org/10.1002/2014JD021665; Dang et al. 2016 JAS, https://doi.org/10.1175/JAS-D-15-0276.1; He et al. 2017 JC, https://doi.org/10.1175/JCLI-D-17-0300.1). I suggest including some discussions on this aspect. Also, did the authors assume spherical snow grains in their model simulations? This should be clarified in the model description.

6. It would be useful if the authors could add a section to discuss the uncertainties

involved in this study for the estimates of aerosol-induced snow albedo effects. For example, uncertainties from overestimated fSCA and SGS, underestimated NSD, AOD biases, etc.

---

## Author Comment (AC1) · 25 Apr 2019

This study performed WRF-Chem high-resolution (~12km) model simulations coupled with the SNICAR model to resolve BC/dust-snow interactions and associated albedo reductions. The model results are evaluated with a suite of observations for snow properties and aerosol conditions. The authors further illustrated the effect of model resolution by comparing with a quasi-global WRF-Chem simulation. This study highlights the importance of model resolution and aerosol-snow interactions in modeling the Tibetan Plateau hydroclimate. The manuscript is generally well-organized and well-written. I have a few comments and suggestions to help improve the manuscript.

Response: We are thankful to the reviewer for the positive comments and helpful suggestions. We have addressed all the suggestions and our point-by-point responses for the specific comments are mentioned below in blue color.

The subsequent modifications and additions in our revised manuscript against each comment are shown in red color.

1. When evaluating modelled LAP concentration in snow, did the authors average the LAP concentrations throughout all the snow layers (up to 5) in the model? Or did the authors only look at specific snow thickness (e.g., top 10 cm) in the model? The observed LAP in surface snow may have different sampling thicknesses/depths. More clarifications could be useful. Similar clarifications should also be provided for snow grain size calculations and evaluation.

Response: In our study, simulated values of LAP concentration and snow grain size from the topmost snow layer are compared to the observed values.

In Figure 7, we show the comparison of simulated annual mean LAP values (circle) and simulated range of midday mean LAP values (box plot) against a range of annual mean LAP measurements from literature (obtained from more than one study over each location). The top layer estimates of LAP in snow depends on meltwater flushing, new snowfall and associated top layer evolution (Flanner et al., 2007, 2012; Oleson et al., 2010). Thus, the snow layer corresponding to LAP concentrations varies daily. At the same time, the corresponding range in measurements (annual mean LAP values) used here (except Pamirs) is in principle representative of different years and different snow depths over a location. We agree with the reviewer that the observed LAPs in surface snow have different sampling depths. To minimize the influence of snow sampling depth variation, we have only used the data from literature which are observed as snow surface measurement or from snow pits having thickness less than 15 cm. Additionally, as our model grid size has spatial resolution of 12 km, it is reasonable to believe that other factors like meteorology conditions, AOD distribution, microphysical parameterizations of LAP-snowmelt association and macrophysics of the simulated snow packs can impose greater biases to our simulated annual mean LAP concentration compared to this discrepancy in snow sample depth. The satellite based observations of snow grain size from

STC-MODSCAG data is representative of the snow surface layer. Therefore, we have used the corresponding model values from top snow layer in snow grain size comparisons.

We have added the following discussions on this uncertainty in the methodology section of revised manuscript and also modified Figure 7 caption for better clarity.

Line numbers: 203-207

Note that the number of snow layers and thickness of top snow surface layer are predicted in the CLM model. Fresh snowfall and melting continuously affect the model surface snow layer thickness (3 cm or less). Therefore, LAP concentrations within each snow layer depend on meltwater flushing, new snowfall and associated top layer evolution (Flanner et al., 2007, 2012; Oleson et al., 2010).

Line number: 376-385

We have used more than one study to have a range of annual mean LAP values over each location so in principle the observations are representative of different years and different snow depths over the same glacier (with an exception over Pamirs). Moreover, simulated annual mean LAP concentration only from the topmost snow surface layer is compared to the observed surface snow LAP in snow concentration, which introduce differences in the snow sample depth used for the evaluation. However, to minimize the influence of snow sample depth variation, we have only used data in the literature which are observed as snow surface measurement or from snow pits having a thickness less than 15 cm.
Line number: 351

Simulated SGS values from the topmost snow surface layer is compared to the MODSCAG SGS retrievals.

2. Introduction (Lines 112-114): For the authors' information, there are a number of valuable recent global modeling studies on the LAP-induced snow albedo effect over the Tibetan Plateau which could be included here as references (e.g., Kopacz et al. 2011 ACP, https://doi.org/10.5194/acp-11-2837-2011; He et al. 2014 GRL, https://doi.org/10.1002/2014GL062191; Zhang et al. 2015 ACP, https://doi.org/10.5194/acp-15-6205-2015).

Response: We thank the reviewer for this information. We have included these and other latest studies in our revised manuscript. The modified text is at Lines 112-117.

Many of these studies used online global model simulations at coarse spatial resolutions of ~50-150 km (Flanner and Zender, 2005; Ming et al., 2008; Qian et al., 2011; Kopacz et al., 2011; Zhang et al., 2015). Other studies employed offline simulation of the snow albedo effect using measured or modelled concentrations of deposited LAP in surface snow or estimated from

atmospheric loading and ice cores (Yasunari et al., 2013; Nair et al., 2013; Wang et al., 2015; He et al., 2014; Santra et al., 2019).

3. Line 343: It is a little weird to identify April-June as the summer season. How about using "pre-monsoon" season?

Response: We have replaced "summer" with "pre-monsoon" in the revised manuscript.

4. Is there any specific reason to select 2013-2014 water year as the simulation period? Is it because the availability of observations?

Response: In the NASA's HMA project, the water year 2013-2014 is identified as a good water year for study. Therefore, we selected this year for satellite retrieval and model simulations.

5. Another important uncertainty factor the authors did not mention is the snow grain shape effect. Recent studies have shown significant effects from nonspherical snow grain shapes on snow albedo and BC/dust-induced albedo reduction (e.g., Liou et al. 2014 JGR, https://doi.org/10.1002/2014JD021665; Dang et al. 2016 JAS, https://doi.org/10.1175/JAS-D-15-0276.1; He et al. 2017 JC, https://doi.org/10.1175/JCLI-D-17-0300.1). I suggest including some discussions on this aspect. Also, did the authors assume spherical snow grains in their model simulations? This should be clarified in the model description.

Response: Yes, we have assumed spherical shaped snow grains in the simulations. In agreement with this suggestion from the reviewer, we have revised our manuscript to include the references and a discussion on the uncertainties in simulated aerosol-induced snow albedo darkening due to snow grain shape assumption.

The modified text is added at Line number 734 in the revised manuscript is shown below.

Moreover, we have assumed spherical shaped snow grains in our simulations. Recently, microscopic level studies show that uncertainties associated with simplified snow grain shape treatment in model parameterization can solely contribute to large biases in SNICAR-$\Delta\alpha$ estimates and thus the LAP-snow albedo radiative and snow melt feedback processes (Liou et al., 2014; Dang et al., 2016; He et al., 2017).

6. It would be useful if the authors could add a section to discuss the uncertainties involved in this study for the estimates of aerosol-induced snow albedo effects. For example, uncertainties from overestimated fSCA and SGS, underestimated NSD, AOD biases, etc

Response: Following this suggestion from the reviewer, we have included a focused discussion on the uncertainties involved in simulated LAP-induced snow albedo darkening.

The modified text is added at Line number 720 in the revised manuscript is shown below.

As already discussed, the simulated fSCA values in WRF-HR are greater than observed fSCA from STC-MODSCAG for most of the winter season (Figure 4E). Specifically, the underestimation in WRF-HR simulated $\Delta\alpha$ values (Figures 9C and 9D) in winter over Karakoram, Hindu Kush and Himalayas is in agreement with corresponding overestimation of WRF-HR simulated fSCA values over these regions (Figure 2). STC-MODDRFS estimated $\Delta\alpha$ is based on surface reflectance, while $\Delta\alpha$ calculated by model involves the surface layer depth. The surface snow layer in SNICAR/CLM continuously evolves as fresh snowfall is added or with snow melting, so the LAP concentrations in the surface layer depend on new snowfall, meltwater flushing, and layer combination/division (Flanner et al., 2007; Flanner et al., 2012; Oleson et al., 2010). Therefore, more precipitation and more snow coverage in winter can be a primary factor causing the underestimation of annual mean LAP concentration and LAP-induced snow darkening. Secondly, the associated overestimation in simulated SGS during winter (Figure 4E) can also contribute to the lower $\Delta\alpha$ values simulated in WRF-HR because bigger snow grains in WRF-HR lead to lower clean albedo and thus smaller percentage reduction in albedo compared to STC-MODDRFS. Moreover, we have assumed spherical shaped snow grains in our simulations. Recently, microscopic level studies show that uncertainties associated with simplified snow grain shape treatment in model parameterization can solely contribute to large biases in SNICAR-$\Delta\alpha$ estimates and thus the LAP-snow albedo radiative and snow melt feedback processes (Liou et al., 2014; Dang et al., 2016; He et al., 2017). Thirdly, the fact that the persistent cloud cover over HMA during winter season can induces a lot of uncertainty in the STC-MODSCAG and STC-MODDRFS estimates, is also equally important.

At the same time, uncertainties regarding aerosol emission, transport and deposition to the snow layers are also significant. It is well known that the transport and deposition of black carbon from Indian landmass to Himalayas increases in the afternoon with the evolution of boundary layer over the IGP region (Dumka et al., 2015;Raatikainen et al., 2014). This feature is well simulated by the model (not shown). As STC-MODDRFS estimates are representative of 1000 LT, but simulated values are sampled in the midday (1000-1400 LT), positive biases in aerosol transport and deposition in snow packs (i.e. higher $\Delta\alpha$ values) might be simulated in WRF-HR runs, especially during pre-monsoon months. At the same time, GOCART dust emission parameterization (used here) is dependent on near surface wind speed. Previous studies have evaluated and illustrated inherent uncertainties in dust emission by this parameterization, mostly underestimation over Indian region (Dipu et al., 2013; Kumar et al., 2014). Thus, the uncertainty in local dust emission fluxes over HMA can also contribute to the biases in simulated $\Delta\alpha$ values. Also, large biases in LAP values may be simulated due to model uncertainties in enhanced wet scavenging fluxes in winter. An overestimation in LAP concentration can lead to an overestimation of snow darkening and melting, resulting in an underestimation of NSD (Figure S4). The large biases in $\Delta\alpha$ values (> 20 %) simulated by WRF-HR

towards late spring could be attributed to both, underestimation in fSCA and overestimation of LAP concentration in the model.

Although a better quantification of these model biases requires evaluation against in-situ measurements, it is worth mentioning here that no in-situ measurements are available for a direct comparison of these high $\Delta\alpha$ values WRF-HR over W. Himalayas (Gertler et al., 2016). Nonetheless, the high values simulated during pre-monsoon are close to previously reported values over other HMA regions. Kaspari et al., (2014) used the offline SNICAR model to report that BC concentrations in pre-monsoon snow/ice samples at Mera Glacier were large enough to reduce albedo by 6-10% and the reduction in albedo was 40-42% relative to clean snow when dust is included in the calculation. Recently, Zhang et al. (2018) has combined a large dataset of LAP measurements in surface snow with the offline SNICAR model to illustrate that $\Delta\alpha$ can be >35% over Tibetan Plateau. Moreover, the composite effect of this discrepancy on seasonal/annual mean values is minimal as the snowpack is at its minimum near the end of pre-monsoon season. Similar high daily variability, huge radiative forcing values (LAPRF ~200 W/m$^2$) and sudden decline in snow depth in late pre-monsoon is also reported over upper Colorado river basin (Skiles et al., 2015; Skiles and Painter, 2017).

---

## Author Comment (AC2) · 25 Apr 2019

**Reviewer#1:**

Based on model simulations and recent satellite observations, the authors examine the skill of high resolution WRF on the impact of snow albedo darkening on radiative forcing. They evaluate the model simulation with various observations. The authors also discussed the benefit of high resolution model, by comparing with coarser resolution model simulation, in aerosol and snowpack distribution over HMA. Lastly, spatial/temporal variation of radiative response are also discussed. This is well organized and clearly written manuscript. I would like to suggest this work for publication subject to a revision to address following concerns.

Response: We are grateful to the reviewer for the thorough reading and insightful comments on our manuscript. We have addressed all the comments and suggestions provided by the reviewer. Our point-by-point responses for the specific comments are mentioned below in blue color.

The subsequent modifications and additions in our revised manuscript are shown in red color.

1) Snow albedo: How was snow albedo estimated when snow cover fraction is less than 1? For example, is albedo difference between WRF-HR and WRF-CR (Fig. 5) due to snow cover fraction or aerosol concentration (LAP)? If snow cover fraction is an important factor, can model snow albedo be compared with STC-MODDRFS?

Response: We thank the reviewer for this suggestion. Simulated annual mean "snow albedo" values are primarily the composite albedo values of snow covered grids. In WRF-Chem-SNICAR simulations, the composite albedo of a snow-covered grid box is computed as weighted average of representative area fractions of sub-grid snow-cover and snow-free regions. Thus, snow cover fraction is indeed a contributor to the differences in the surface albedo estimations between the two simulations. However, the differences in the annual/seasonal mean snow albedo values (between WRF-HR and WRF-CR) is resultant of differences in several other factors also i.e. the snow grain size and associated evolution, LAP concentration and snow duration. As discussed in the manuscript, better representation of terrain elevation in WRF-HR (via high spatial resolution) run brought synergistic improvement in all these factors and improves the simulation of albedo over snow-covered grids.

We have included the following discussion about contribution of snow fraction in our comparison of Figure 5 (snow albedo) in the revised manuscript at Line nos 537.

The improvement in α estimation from WRF-HR compared to WRF-CR can be attributed to the relatively better simulation of fSCA (Figure 2), NSD (Figure 3) and SGS (Figure 4). Simulated annual mean α values are primarily the composite albedo values of snow covered grids. In WRF-Chem-SNICAR simulations, the composite albedo of a snow-covered grid box is computed as weighted average of representative area fractions of sub-grid snow-cover and snow-free regions. Thus, relatively lower values of simulated fSCA and NSD in WRF-CR runs

compared to WRF-HR runs can contribute substantially to the relatively lower annual and seasonal mean α values simulated by WRF-CR.

At the same time, the mean α values from STC-MODSCAG (representative of only snow covered regions) are biases towards higher values than the corresponding simulated α values (composite albedo of the pixel) predominantly over the snow grids with annual mean fSCA are much smaller than 1.

2) Season: Winter and summer are defined as DJF and AMJ in main text while NDJF and MAMJ are used in supplementary material. Spring is more appropriate for AMJ as well as MAMJ.

Response: We have replaced "summer" with "pre-monsoon" in the revised manuscript and have also corrected the information in supplementary material.

3) Line 336: 16:00LT → 14:00LT

Response: We have corrected this mistake.

4) Validation of LAP (Lines362-370): What would be the effect of using LAP data observed in different year and different glaciers?

Response: We have added the following sentences in revised manuscript at Line numbers 373-380 to discuss this issue.

We have used more than one study to have a range of annual mean LAP values over each location so in principle the observations are representative of different years and different snow depths over the same glacier (with an exception over Pamirs). Moreover, simulated annual mean LAP concentration only from the topmost snow surface layer is compared to the observed surface snow LAP in snow concentration, which introduce differences in the snow sample depth used for the evaluation. However, to minimize the influence of snow sample depth variation, we have only used data in the literature which are observed as snow surface measurement or from snow pits having a thickness less than 15 cm.

5) Lines 416-418: It seems obvious, but any thought why model overestimate snow cover fraction in DJF and underestimate in AMJ? Does model simulate reasonable surface temperature and precipitation over the region?

Response: We thank the reviewer for this suggestion. We analyzed the simulated spatial and temporal variation in surface rainfall (as a proxy for surface precipitation) and found that WRF-HR indeed overestimates precipitation, which can contribute to the overestimation of simulated fSCA as well as affect the surface temperature. Accordingly, we have added the following figure in supplementary and sentences at Line number 432-435 in the revised manuscript.

Further, WRF-HR simulated annual surface rainfall in winter is overestimated over Karakoram, Himalayan and Hindukush ranges (Figure S3). This indicates that overestimation of surface precipitation in WRF-HR may also contribute to the overestimation of fSCA over HMA in winter.

[Figure]

Figure S3: Spatial distribution of annual mean surface precipitation (in mm/day) for water year 2013-14 from A) TRMM satellite, B) WRF-HR and C) WRF-CR simulations are shown. Corresponding Time-Longitude distribution of latitudinal-mean rainfall is also shown in Panels D-F, respectively.

6) Line 461: Is "and 4F" typo?

Response: Yes, we have corrected this mistake.

7) Lines 498-503: Hard to read. Are "13" and "14" day of the month?

Response: Here, 13 and 14 meant 2013 and 2014. We have revised the sentences accordingly to replace "13" and "14" with "2013" and "2014", respectively, in that paragraph for clarity.

8) Lines 509-512: It's hard to follow the argument. Would you get the same conclusion if a different grid is used?

Response: Figures 4 show that snow grain size (SGS) values over the HMA region are better simulated in WRF-HR runs compared to WRF-CR runs. An anomaly to this feature is seen over a few grids in western Himalaya, where, the WRF-CR simulated SGS values are closer to observations than that simulated in WRF-HR. The fSCA-SGS association over these specific grids is discussed in Figure 4E to understand this anomaly. We illustrate that the model biases in

simulated SGS values are mainly proportional to the corresponding biases in fSCA and NSD, which is valid for all grids across the HMA region.

We have rephrased the paragraph at Line numbers 509-527 as below for more clarity.

Quite anomalous to other grids in HMA, the WRF-CR simulated SGS values over some grids near the Chotta Shingri glacier (marked by magenta circle in Figure 4A) of the western Himalaya sub-region are closer to STC-MODSCAG observations than that simulated by WRF-HR runs. As a sanity check, daily variation of SGS (hashed lines in Figure 4E) and fSCA (solid lines in Figure 4E) from STC-MODSCAG (black), WRF-HR (blue) and WRF-CR (red) over this glacier are compared. Fractional snow cover from STC-MODSCAG gradually increase from 0.2 in November, 2013 to 0.8 in February, 2014 and subsequently decrease back to 0.1 by September, 2014 at the glacier location.  Corresponding SGS values from STC-MODSCAG closely followed the seasonal trend in fSCA varying around the values of 80-200 micron in winter. In comparison, simulated fSCA from WRF-HR values drastically increased to 1 at the beginning of November, 2013 (from no snow cover before that), remained fully snow covered till mid-June, 2014 and then suddenly became snow free after June. Compared to satellite estimates, fSCA from WRF-HR are greater in magnitude throughout the duration of snow cover indicating more snow mass simulated by WRF-HR. Associated SGS values (80-800 micron) simulated by WRF-HR are also greater than STC-MODSCAG estimates throughout the snow duration over the grid. However, the fSCA variation from WRF-CR over this grid is very close to the variation seen by STC-MODSCAG, and the associated SGS values (50-400 micron) from WRF-CR are also closer to the estimated STC-MODSCAG values, supporting our argument that biases in simulation of fSCA also affect the simulated annual mean SGS values.

9) Line 515: Fig. 4 → Fig. 5

Response: Yes, we have corrected this mistake.

10) Fig. 4 has a wrong caption

Response: Yes, we have corrected this mistake.

11) Fig. 5 has a wrong caption

Response: Yes, we have corrected this mistake.

12) Fig. 7A: What cause the high LAP in the south of Taklamakan desert?

Response: Local dust emissions and dust transported from Taklimakan desert is the source of LAP concentration in snow over the HMA ranges to the south of Taklimakan desert. Moreover, improvement in fractional snow cover (Figure 2B and 2C) and snow duration (Figure 3B and 3C) over the entire stretch of HMA ranges to the south of Taklimakan desert is also seen in WRF-HR runs (compared to WRF-CR runs). Thus, longer duration of snow cover and greater dust

deposition in WRF-HR (Figures S6, S7) result in higher annual mean LAP values than WRF-CR runs.

13) Spatial pattern of LAP (Fig. 7) doesn't seem to highly correlated with albedo (Fig. 5)

Response: Annual mean snow albedo spatial pattern is primarily governed by spatial pattern and frequency of snowfall, snow grain size evolution and associated aging, LAP concentration and the snow pack duration. In contrast, the annual mean spatial pattern of LAP concentration in snow is governed by aerosol emission, transport and meteorology/terrain induced deposition of aerosols to the snow packs and snow cover duration. Inherent differences in these governing factors results in these two variables not being highly correlated spatially over HMA. Secondly, the LAP concentration is higher at lower elevations, but comparatively snow albedo values are greater at higher elevations. Thirdly, more LAP concentration itself decreases the snow albedo values, making these two variables anti-correlated within the same grid.

14) AOD is higher in CR, but LAP is higher in HR. Yet, albedo is higher in HR in general. It is probably due to higher snow cover (See also my comment #1) and more snowfall (Lines 654-657) in HR. If so, additional simulation (with similar climatology) may be required to discuss the impact of snow darkening.

Response: AOD is the measure of particles airborne in model atmosphere layers. LAP in snow is the measure of particles deposited in the surface snow layer of the model. The lifetime of airborne aerosols and deposition of airborne aerosols to snow packs is dependent on precipitation (wet deposition) and terrain (dry deposition). Greater snowfall and greater terrain heights over HMA are simulated in WRF-HR (compared to coarser resolution runs), which enhances the total deposition flux of aerosols into snow packs. This results in relatively higher LAP in snow surface (but lower lifetime and AOD) over HMA than that simulated in WRF-CR (Figures 7 and 8 and Figures S6-S7). As replied to comment #1, the differences in simulation of many factors i.e. aerosol emissions, transport, meteorology, fSCA, and snow microphysical properties all contributes to the differences in AOD and LAP better WRF-HR and WRF-CR.

The differences in LAP-induced snow albedo darkening (between WRF-HR and WRF-CR) is primarily governed by the LAP concentration in top snow layer. Larger LAP concentration can induce more snow darkening. The comparisons presented in our manuscript clearly show the improvement in WRF-HR simulated LAP concentration in snow and thus argue that higher spatial resolution will bring improvement in simulation of LAP-induced snow albedo darkening. Considering that meteorology itself, including precipitation, is coupled to the terrain elevation over HMA, meteorological conditions seldom remains constant (to manifest similar climatology) while studying the effect of spatial resolution on aerosol-snow interactions. Our study is meant to study such effect of model resolution in the coupled atmosphere-snow-terrain system with all the interactions considered in a consistent manner.

15) Fig. 9C & Lines 866-868: What cause the bias? Does model has local source of dust over HMA?

Response: The biases in WRF-HR simulated LAP-induced snow darkening in winter in comparison to the STC-MODDRFS estimates (Fig. 9C and Lines 866-868) is mainly attributable to uncertainty in simulation of snow layer properties (fSCA, NSD, SGS), aerosol emission, distribution, transport and deposition (AOD, LAP, meteorology) and various inherent assumptions within the model parameterization towards representation of snow morphology and LAP-snowmelt dynamical feedbacks in our model.

Yes, our model has local sources of dust over HMA (see Figure R1 in comment #16). Following Zhoa et al., 2010, the dust emission scheme adopted from the Goddard Chemistry Aerosol Radiation and Transport (GOCART) model (Ginoux et al., 2001) coupled with the MOSAIC aerosol schemes is used in this study. GOCART dust emission parameterization is dependent on near surface wind speed. Previous studies have evaluated this parameterization and illustrated inherent uncertainties, mostly underestimation over Indian region (Dipu et al., 2013; Kumar et al., 2014). Thus, the uncertainty in local dust emissions (along with other factors) can also contribute to the biases in simulated Δα values. We thank the reviewer for directing us towards this uncertainty.

We thank the reviewer for this comment about local dust emissions. We have included a discussion focused on all the uncertainties involved in simulated LAP-induced snow darkenning in the revised manuscript at Line numbers 719-771.

As already discussed, the simulated fSCA values in WRF-HR are greater than observed fSCA from STC-MODSCAG for most of the winter season (Figure 4E). Specifically, the underestimation in WRF-HR simulated Δα values (Figures 9C and 9D) in winter over Karakoram, Hindu Kush and Himalayas is in agreement with corresponding overestimation of WRF-HR simulated fSCA values over these regions (Figure 2). STC-MODDRFS estimated Δα is based on surface reflectance, while Δα calculated by model involves the surface layer depth. The surface snow layer in SNICAR/CLM continuously evolves as fresh snowfall is added or with snow melting, so the LAP concentrations in the surface layer depend on new snowfall, meltwater flushing, and layer combination/division (Flanner et al., 2007; Flanner et al., 2012; Oleson et al., 2010). Therefore, more precipitation and more snow coverage in winter can be a primary factor causing the underestimation of annual mean LAP concentration and LAP-induced snow darkening. Secondly, the associated overestimation in simulated SGS during winter (Figure 4E) can also contribute to the lower Δα values simulated in WRF-HR because bigger snow grains in WRF-HR lead to lower clean albedo and thus smaller percentage reduction in albedo compared to STC-MODDRFS. Moreover, we have assumed spherical shaped snow grains in our simulations. Recently, microscopic level studies show that uncertainties associated with simplified snow grain shape treatment in model parameterization can solely contribute to large biases in SNICAR-Δα estimates and thus the LAP-snow albedo radiative and snow melt feedback processes (Liou et al., 2014; Dang et al., 2016; He et al., 2017). Thirdly, the fact that the

persistent cloud cover over HMA during winter season can induces a lot of uncertainty in the STC-MODSCAG and STC-MODDRFS estimates, is also equally important.

At the same time, uncertainties regarding aerosol emission, transport and deposition to the snow layers are also significant. It is well known that the transport and deposition of black carbon from Indian landmass to Himalayas increases in the afternoon with the evolution of boundary layer over the IGP region (Dumka et al., 2015;Raatikainen et al., 2014). This feature is well simulated by the model (not shown). As STC-MODDRFS estimates are representative of 1000 LT, but simulated values are sampled in the midday (1000-1400 LT), positive biases in aerosol transport and deposition in snow packs (i.e. higher Δα values) might be simulated in WRF-HR runs, especially during pre-monsoon months. At the same time, GOCART dust emission parameterization (used here) is dependent on near surface wind speed. Previous studies have evaluated and illustrated inherent uncertainties in dust emission by this parameterization, mostly underestimation over Indian region (Dipu et al., 2013; Kumar et al., 2014). Thus, the uncertainty in local dust emission fluxes over HMA can also contribute to the biases in simulated Δα values. Also, large biases in LAP values may be simulated due to model uncertainties in enhanced wet scavenging fluxes in winter. An overestimation in LAP concentration can lead to an overestimation of snow darkening and melting, resulting in an underestimation of NSD (Figure S4). The large biases in Δα values (> 20 %) simulated by WRF-HR towards late spring could be attributed to both, underestimation in fSCA and overestimation of LAP concentration in the model.

Although a better quantification of these model biases requires evaluation against in-situ measurements, it is worth mentioning here that no in-situ measurements are available for a direct comparison of these high Δα values WRF-HR over W. Himalayas (Gertler et al., 2016). Nonetheless, the high values simulated during pre-monsoon are close to previously reported values over other HMA regions. Kaspari et al., (2014) used the offline SNICAR model to report that BC concentrations in pre-monsoon snow/ice samples at Mera Glacier were large enough to reduce albedo by 6-10% and the reduction in albedo was 40-42% relative to clean snow when dust is included in the calculation. Recently, Zhang et al. (2018) has combined a large dataset of LAP measurements in surface snow with the offline SNICAR model to illustrate that Δα can be >35% over Tibetan Plateau. Moreover, the composite effect of this discrepancy on seasonal/annual mean values is minimal as the snowpack is at its minimum near the end of pre-monsoon season. Similar high daily variability, huge radiative forcing values (LAPRF ~200 W/m$^2$) and sudden decline in snow depth in late pre-monsoon is also reported over upper Colorado river basin (Skiles et al., 2015; Skiles and Painter, 2017)

16) Same emission databases on relatively coarse resolution are used for both simulations. How about actual emission of dust and BC? Are they comparable in HR and CR?

Respond: We agree with the reviewer that there is uncertainty in the emission rates we used in our simulations. Real-world fugitive emissions are still not accounted in coarsely resolved inventories which can indeed induce uncertainty in our simulations. Moreover, scarcity in

three-dimensional measurements have limited certainty analysis of various global inventories over Asia including HMA region. Nonetheless, the main objective of our study is to illustrate the impact of horizontal resolution on aerosol, LAP and snow properties via improved terrain representation, so we have kept the emission inventories same for both the runs, WRF-HR and WRF-CR. Although, dust emissions in both HR and CR runs vary as it depends on simulated meteorology (Figure R1), the emission fluxes are comparable over HMA. The black carbon emission fluxes used in both HR and CR runs are also comparable (same anthropogenic emission inventory).

We included the following sentence in the revised manuscript at Line number 594.

Moreover, Jayarathne et al., 2018 shows that many local emissions are not accounted in coarse emissions which causes underestimation in simulated regional AOD values in these valleys.

[Figure]

Figure R1: Spatial distribution of mean deposition rate for A) WRF-CR runs and B) WRF-HR during winter months. Panel C) and D) are same as Panels A) and B), but, for pre-monsoon months.